# iNOS is essential to maintain a protective Th1/Th2 response and the production of cytokines/chemokines against *Schistosoma japonicum* infection in rats

Jia Shen[1,2], Si-fei Yu[3], Mei Peng[1,2], De-Hua Lai[4], Geoff Hide[5], Zhong-Dao Wu[1,2]*, Zhao-Rong Lun[4]*

**1** Department of Parasitology and Key Laboratory of Tropical Disease Control (SYSU), Ministry of Education, Zhongshan School of Medicine, Sun Yat-Sen University, Guangzhou, P.R. China, **2** Provincial Engineering Technology Research Center for Biological Vector Control, Guangzhou, Guangdong, P.R. China, **3** Clinical Research Institute, The First People's Hospital of Foshan, Foshan, P.R. China, **4** State Key Laboratory of Biocontrol, School of Life Sciences, Sun Yat-Sen University, Guangzhou, P.R. China, **5** Ecosystems and Environment Research Centre and Biomedical Research Centre, School of Science, Engineering and Environment, University of Salford, Salford, United Kingdom

* wuzhd@mail.sysu.edu.cn (Z-DW); lsslzr@mail.sysu.edu.cn (Z-RL)

**Data Availability Statement:** All relevant data are within the manuscript and its Supporting Information files.

## Abstract

Humans and a wide range of mammals are generally susceptible to *Schistosoma* infection, while some rodents such as *Rattus* rats and *Microtus* spp are not. We previously demonstrated that inherent high expression levels of nitric oxide (NO), produced by inducible nitric oxide synthase (iNOS), plays an important role in blocking the growth and development of *Schistosoma japonicum* in wild-type rats. However, the potential regulatory effects of NO on the immune system and immune response to *S. japonicum* infection in rats are still unknown. In this study, we used iNOS-knockout (KO) rats to determine the role of iNOS-derived NO in the immune system and immunopathological responses to *S. japonicum* infection in rats. Our data showed that iNOS deficiency led to weakened immune activity against *S. japonicum* infection. This was characterized by the impaired T cell responses and a significant decrease in *S. japonicum*-elicited Th2/Th1 responses and cytokine and chemokine-producing capability in the infected iNOS-KO rats. Unlike iNOS-KO mice, Th1-associated cytokines were also decreased in the absence of iNOS in rats. In addition, a profile of pro-inflammatory and pro-fibrogenic cytokines was detected in serum associated with iNOS deficiency. The alterations in immune responses and cytokine patterns were correlated with a slower clearance of parasites, exacerbated granuloma formation, and fibrosis following *S. japonicum* infection in iNOS-KO rats. Furthermore, we have provided direct evidence that high levels of NO in rats can promote the development of pulmonary fibrosis induced by egg antigens of *S. japonicum*, but not inflammation, which was negatively correlated with the expression of TGF-β3. These studies are the first description of the immunological and pathological profiles in iNOS-KO rats infected with *S. japonicum* and demonstrate key differences between the responses found in mice. Our results significantly enhance our

**Funding:** This work was supported by grants from the National Natural Science Foundation of China (Grant No. 81802036, J.S.; No. 81871682, Z.D.W.; http://www.nsfc.gov.cn/), the National Key R&D Program of China (No. 2020YFC1200100 and No. 2016YFC1200500, Z.D.W.; https://program.most. gov.cn/), the Natural Science Foundation of Guangdong Province, China (2020A1515010896, J.S.; http://pro.gdstc.gd.gov.cn/), the China Postdoctoral Science Foundation (No. 2018M631027 and 2019T120770, J.S.; http://jj. chinapostdoctor.org.cn/). The funders had no role in study design, data collection and analysis, decision to publish, or preparation of the manuscript.

**Competing interests:** The authors have declared that no competing interests exist.

understanding of the immunoregulatory effects of NO on defensive and immunopathological responses in rats and the broader nature of resistance to pathogens such as *S. japonicum*.

## Author summary

Schistosomiasis is a zoonosis that affects more than 200 million people worldwide. A wide range of mammals, including mice, are permissive hosts of *Schistosoma* and develop chronic disease characterized by egg-granuloma formation and fibrosis after infection. Rats, on the other hand, are non-permissive hosts and develop efficient immune responses to eliminate the worms. Interestingly, schistosome eggs elicit a dominant Th2 immune response within mouse hosts, whereas rats with schistosomiasis develop a significant Th2 response in the absence of available egg production. The Th2 response in rats seems to play an essential role in the protection of the host against *Schistosoma*. So far, the factors that lead to the different immune responses to *Schistosoma* infection in both hosts have not been demonstrated. In this study, our results show that an iNOS-dependent mechanism maintains the function of the immune system in rats by modulating $CD4^+$ T cell-mediated Th1/Th2-associated cytokine responses and chemokine production. Additionally, the absence of iNOS led to slow clearance of parasites, increases in the development of worms, and an exacerbation of granuloma formation and fibrosis in rats. Furthermore, high levels of NO in rats can promote the development of fibrosis induced by inflammation (rapid inflammatory repair). Therefore, this study demonstrates that the difference in iNOS levels between mice and rats is responsible for the different immune responses and outcomes induced by schistosome infection in both hosts.

## Introduction

Schistosomiasis is a zoonosis caused by the parasitic helminth *Schistosoma*. It is an important public health problem that affects around 290 million people in 78 countries [1]. The bulk of the morbidity and mortality of schistosomiasis ultimately results from the development of a Th2-driven inflammatory response and fibrosis in the host caused by *Schistosoma* eggs trapping in tissues, such as the intestines and liver (intestinal schistosomiasis) and the urinary bladder wall (urogenital schistosomiasis) [2]. A wide range of mammals, including humans, are generally susceptible/permissive hosts of *Schistosoma* and develop chronic disease characterized by egg-granuloma formation and fibrosis after infection [3,4]. In contrast, the rat is a nonpermissive host in which the parasite does not cause typical egg granulomas in the liver of the infected host. This is because the rat is genetically resistant to *Schistosoma* infection and develops efficient immune responses to eliminate the worms [4,5]. Typically, mice are widely used as a model of a permissive host, for *Schistosoma* spp, to understand the immune responses during human schistosomiasis. During the initial stages of infection, mice display a Th1 immune response by producing high levels of Th1 cytokines (e.g., IFN-γ, interleukin (IL) -12, TNF-α), which may participate in protection against *Schistosoma* infection [6]. However, once *Schistosoma* eggs are produced at 5–6 weeks of infection, the host immune status dramatically shifts to a Th2 response, as shown by the increased production of Th2-associated cytokines IL-4, IL-5, IL-10, and IL-13 [7,8]. The Th2 response is involved in the development of egg-granuloma formation and fibrosis and is essential for host survival by protecting against inflammatory cytokines that mediate death during acute schistosomiasis. The excretory-

secretory antigens (ES) from the eggs are powerful factors that induce Th2 polarization in schistosomiasis. A particularly striking difference, in the rat model of schistosomiasis compared to mice, is the fact that a Th2 response is driven in the absence of available egg production by worm pairs [4,9]. Therefore, it suggests that the, significantly greater, Th2 type response in rats seems to play an essential role in the protection of the host against *Schistosoma* [4]. Consequently, the different immune responses induced by schistosomes in mice and rats may be responsible for the different outcomes of host adaptability. However, the factors that lead to the different immune responses to *Schistosoma* infection in both hosts have not been elucidated. In previous work, we demonstrated a key role of nitric oxide in resistance to schistosome infection in rats suggesting its involvement in this process [10].

Nitric oxide (NO), produced from L-arginine by three isoforms of nitric oxide synthase (NOS), exhibits a variety of physiologic functions in mammals [11]. It is reported that NO plays an important role in regulating vascular function, neurotransmission, inflammatory responses, immune function, and host defense [12,13]. Inducible NOS (iNOS), expressed in response to proinflammatory cytokines (such as IFN-γ, TNF-a, and IL-1β) and/or microbial products (such as LPS), can rapidly produce large amounts of NO in contrast to the other two isoforms, the endothelial NOS (eNOS) and the neuronal NOS (nNOS) [12]. In previous studies, we have confirmed that the expression of iNOS and the levels of NO induced by stimulation of macrophages of rats are significantly higher than those in mice [10,14,15]. Furthermore, using iNOS knockout (iNOS-KO) rats, we have demonstrated that NO, inherently at high levels in rats, plays a crucial role in blocking *S. japonicum* growth, reproductive organ development, egg production, and the ability to lay fertilized eggs [10]. Furthermore, the inhibitory effect of NO acts by affecting mitochondrial respiration and energy production in this parasite [10]. Indeed, in addition to its direct cytotoxic effect on infectious pathogens in non-specific immune defense mechanisms, NO has also been found to be a potent immunoregulatory factor [16,17]. Previously, studies have shown that NO displays a significant immunosuppressive effect by causing inhibition of T cell proliferation, regulation of T-cell function [18–20], and inducing T helper cell deviation by suppression of Th1 (and Th2) cell responses [21,22]. Moreover, NO has also been demonstrated to influence the expression and function of many inflammatory factors [23]. Therefore, it raises the question as to whether the high expression of NO in rats affects the development of *S. japonicum* by regulating the immune response.

The majority of previous studies on the effects of NO on the immune system were based on mouse models. However, the effects of NO on the immune system vary with concentration, host species, and response to different pathogens. For example, it was reported that high concentrations of NO could suppress the expansion of Th1 cells by inhibiting IL-12 synthesis, whereas low doses of NO selectively promoted Th1 cell differentiation and had no effect on Th2 cells [21]. Furthermore, NO has been reported to preferentially down-regulate Th1-mediated immune responses in the murine system [24,25], however, this selective inhibitory effect on Th1 responses was not found in activated human T cells [22]. Additionally, NO induces different immune responses to different pathogens in the same host. It is shown that a Th1 immune response was enhanced in iNOS-KO mice after infection with *Leishmania major* [26], whereas a gastric *Helicobacter* infection displayed a reduction of the Th1 response in iNOS-KO mice, relative to that of the WT mice [27]. Thus far, little is known in rats about the regulatory role of NO on the immune system and the outcomes of iNOS gene deficiency on the evolving immune response. The differences in NO levels between mice and rats [14] may be linked to differences in *Schistosoma* infection-induced immune responses. Therefore, there is a need to investigate the role of NO in the regulation of the immune response that inhibits *Schistosoma* infection in rats.

In the present study, using *S. japonicum* infected iNOS-KO and wild-type rat models, we investigated the effects of iNOS-derived NO on immunity and immunopathological responses to *S. japonicum* infection in rats. Our results show that an iNOS-dependent mechanism maintains the function of the immune system in rats by modulating CD4$^+$ T cell-mediated Th1/Th2-associated cytokine responses and chemokine production. This plays a crucial role in host protective immunity against *S. japonicum* infection in rats. Thus, this study significantly enhances our understanding of the immunoregulatory effects of NO on defensive and immunopathological responses in rats.

## Results

### iNOS-dependent functions are essential for host protection against *S. japonicum* infection in rats

To examine the role of iNOS against *S. japonicum* infection in rats, WT and iNOS-KO rats were infected with *S. japonicum*. As shown in Fig 1A, the worm burden collected from the hepatic portal vein infusion of infected KO rats, compared with infected WT rats, was greater by about 1.5-fold at week 1 (*P*<0.05) and 3-fold at week 7 post-infection (*P*<0.01). Interestingly, although the worm load showed a slight increase in the infected KO rats at week 4 post-infection and was higher than those found at week 1 post-infection, there was no significant difference (*P*>0.05) between the two groups. The results suggest that iNOS in rats promotes the elimination of *S. japonicum*, particularly after 4 weeks post-infection. Additionally, when we compared the development of *S. japonicum* in the WT and iNOS-KO rats, the results of SEM showed that most of the spines in the middle part of the body of the schistosomula from iNOS-KO rats had disappeared. This is similar to what is observed in infected mice, while large numbers of spines on the tegument were still present in the middle part of the body of the schistosomula from WT rats (S1 Fig). In the adult stages (7 weeks after infection), the development of *S. japonicum* in iNOS-KO rats is markedly more healthy than those in WT rats, which are larger and longer (Fig 1B).

Hepatic granulomatous inflammation and fibrosis is the primary cause of chronic morbidity in schistosomiasis. Strikingly, the infected KO rats developed significant granulomatous inflammation and fibrosis in the liver, particularly at the chronic time point of 12 weeks post-infection, compared with infected WT rats (Fig 1C). Dead worms in the liver sections were observed in the infected KO rats (Fig 1C). Moreover, infected KO rats experienced significantly greater weight loss than infected WT rats, especially after 4 weeks post-infection (*P*<0.01) (Fig 1D). When infected with a low dose of 200 cercariae of *S. japonicum*, no death was found in either the WT or KO rats. However, at high doses (1000 cercariae per rat), the survival of iNOS-KO rats was dramatically decreased and ultimately 100% of animals succumbed between days 11–34 post-infection, while no mortality was observed in the WT control group during this period (Fig 1E). Together, these results showed that iNOS-mediated functions are essential in host protection against *S. japonicum* infection in rats, as characterized by accelerating clearance of parasites, alleviating hepatic pathological responses, and promoting host survival.

### iNOS deficiency led to a decrease in the frequency of T cells and an increase in B cells in rats infected with *S. japonicum*

In order to determine whether iNOS is involved in anti-*Schistosoma* infective immunity by affecting the profile of immune cells in rats, we examined the differences in frequencies of T cells and B cells in spleens, mesenteric lymph nodes (LN), and liver cells by comparing infected

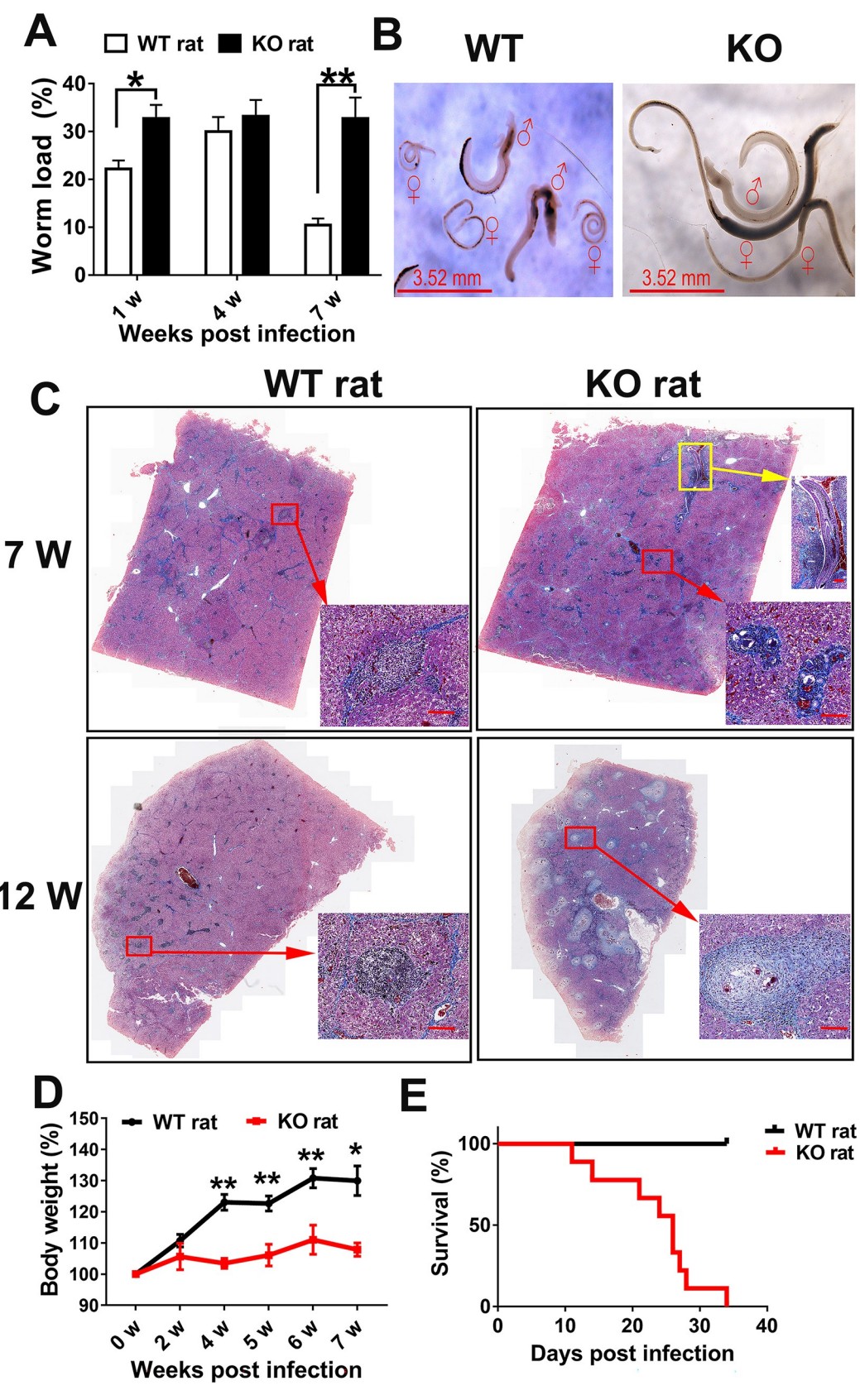

**Fig 1. iNOS-KO rats show slow clearance of *S. japonicum*, exacerbated pathological responses and increased morbidity.** Wildtype (WT) and iNOS knock-out (KO) rats were infected with 200 *S. japonicum* cercariae, parasites were perfused from the portal tract at 1, 4- and 7-weeks post-infection and counted under a stereoscopic microscope. (A) Percentage worm load collected from hepatic portal vein infusion. The data are expressed as the mean ± SEM of 5 rats per group. * $P <0.05$, ** $P <0.01$. (B) Micrographs of representative parasites from WT and iNOS-KO rats. ♂, male worms; ♀, female worms. Bar = 3.52mm. (C) Histopathology of *S. japonicum* infected WT and iNOS-KO rats. Liver sections were stained with Masson's Trichrome at 7- and 12-weeks post-infection. Blue (aniline blue) represents collagen fibers. Red arrows indicate the granulomas, and the yellow arrow indicates the worm. Bar = 100 um. (D) Weight change throughout 7 weeks period. n = 5 per group. The data are expressed as the mean ± SEM of 5 rats per group. * $P <0.05$, ** $P <0.01$. (E) Percentage survival of WT and iNOS-KO rats following a high dose (1000 cercariae) infection of *S. japonicum*. n = 9 per group.

iNOS-KO rats and WT rats. In comparison with WT rats, the results showed that there was a lower frequency of CD3+ T cells ($P<0.05$) and a significantly higher frequency of CD19+ B cells ($P<0.05$) both in LN and liver cells of KO rats (naïve and infected) but not in spleens (Fig 2A and 2B). However, no significant differences were observed in the frequency of macrophages in the spleen, LN, and liver between WT rats and KO rats ($P>0.05$) (S2 Fig). Thus, these results suggested the role of iNOS in maintaining the host's T-cell immune response and the immunosuppressive effect on B cells in rats during *S. japonicum*. Furthermore, it indicates that iNOS-mediated T-cell immune responses may play an important role in host protection against *S. japonicum* infection in rats.

To evaluate the role of iNOS in regulating the host's T cell immune function in rats during *Schistosoma* infection, the proportion of CD4+ and CD8+ T cell subsets in spleens, mesenteric LN, and livers from WT and iNOS-KO rats were analyzed. As shown in Fig 2C and 2D, the naïve KO rat displayed a lower frequency of CD4+ T cells in the LN ($P<0.05$) in comparison with naïve WT rats. However, there was no significant difference in the percentage of CD8+ T cells in all examined tissues between the two naïve groups. After infection with *S. japonicum*, WT rats showed a decrease in the percentage of CD4+ T cells in the spleen ($P<0.05$), LN ($P<0.05$), and liver ($P<0.01$), and a slight increase in the percentage of CD8+ T cells in the liver ($P<0.05$). Surprisingly, an increase in the percentage of CD4+ T cells and a reduction in the percentage of CD8+ T cells in the spleen were found in the KO rats following infection ($P<0.05$). Interestingly, the results found for the liver were opposite to the other tissues.

## A profile of pro-inflammatory and pro-fibrogenic cytokines developed in iNOS deficient rats with schistosomiasis

To investigate the systemic immune response developed in KO rats, cytokine-producing profiles were measured in serum at different time points post-infection with *S. japonicum*. These included the pro-inflammatory cytokines (TNF-α, IL-6, and IL-1β), Th1 cytokines (IFN-γ, IL-2), Th2 cytokines (IL-4, IL-5, and IL-13), and regulatory cytokines (IL-10). As demonstrated in Fig 3, when the iNOS gene was ablated in rats, it developed a pro-inflammatory immune environment and Th1 immune deviation, with an increase in pro-inflammatory cytokines (TNF-α, IL-1β, and IL-6, $P<0.05$) (Fig 3A) and in Th1 cytokines (IFN-γ, $P<0.05$) (Fig 3B) in naïve KO rats in comparison with naïve WT rats (0 w post-infection). After infection with *S. japonicum*, KO rats also developed a strong pro-inflammatory response, with high production of TNF-α, IL-6, and IL-1β at all time points post-infection which reached a peak at 7 weeks post-infection. Moreover, infected KO rats displayed a marked decrease in the levels of both Th1- (IFN-γ) and Th2-associated cytokines (IL-4, IL-5) at all time points post-infection, except with IL-2 and IL-13 (Fig 3C), both of which play an important role in granuloma and fibrosis development [26,27]. In addition, the immunosuppressive cytokine IL-10 was also significantly down-regulated in KO rats, which displayed a notable significant difference at 7 weeks

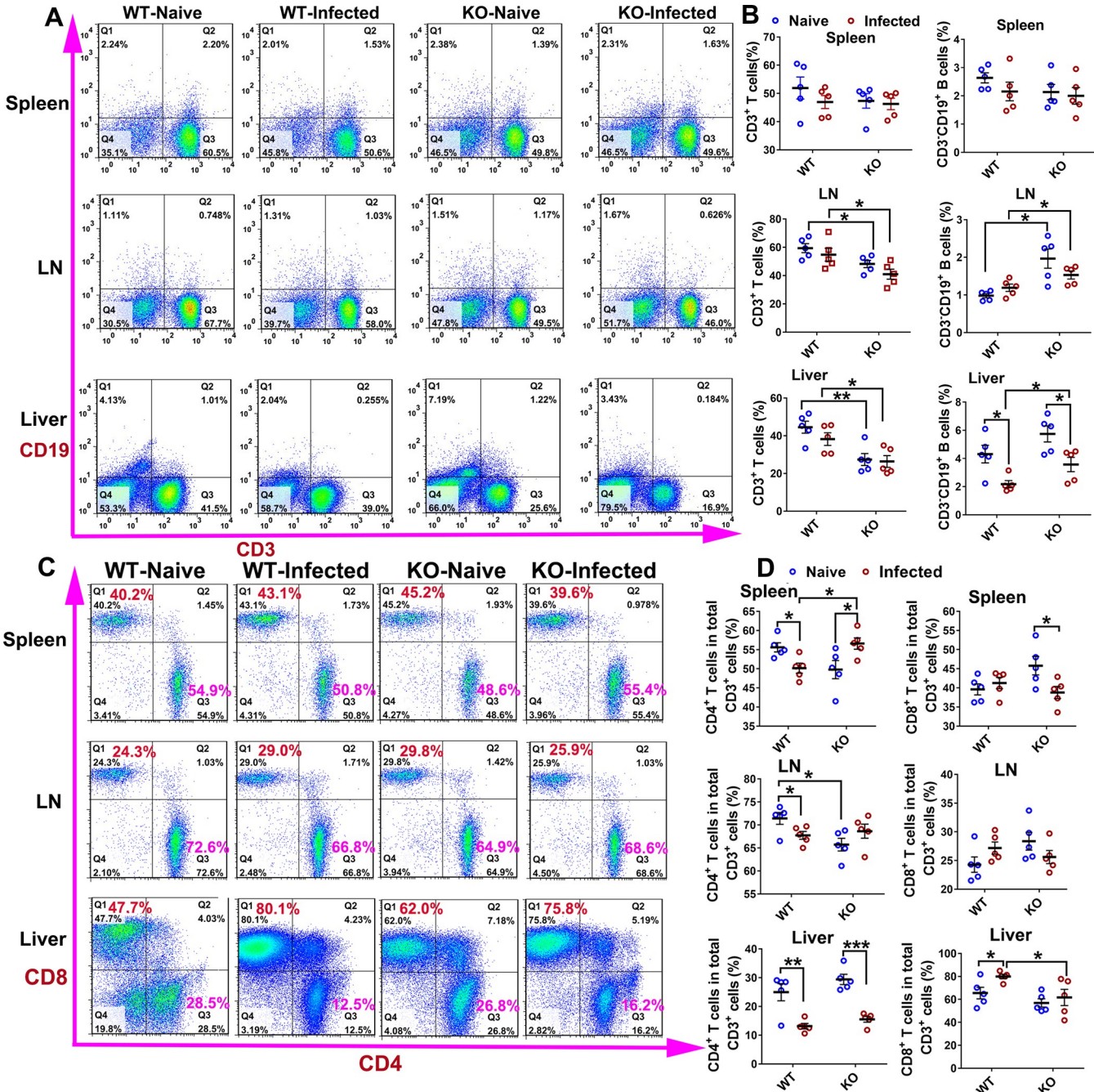

**Fig 2. Flow cytometry analysis of the changes in T cell and B cell proportions in iNOS-KO and WT control rats.** WT and KO rats were infected percutaneously with 200 *S. japonicum* cercariae and sacrificed at 7 weeks post-infection. (A) Representative FACS plots of CD3$^+$ T cells and CD19$^+$ B cells in the spleens, LN, and livers. (B) Frequency of CD3$^+$ T cells and CD19$^+$ B cells in the spleens, LN, and livers. (C) Representative FACS plots of CD4$^+$ and CD8$^+$ T cells in the spleens, LN, and livers. Numbers represent the frequency of the boxed population within the CD3$^+$ T cell population. (D) Frequency of CD4$^+$ and CD8$^+$ T cells. Data shown are means ± SEM and repeated twice with similar results, n = 5 rats per group. Significant differences have been noted, $^*$ $P<0.05$, $^{**}P<0.01$, $^{***}P<0.001$. SP = Spleen, LN = lymph node.

post-infection between both groups (Fig 3D). Expression of IL-17 was not detectable. Thus, it suggests that a pro-inflammatory and pro-fibrogenic immunological environment developed with iNOS deficiency in infected rats.

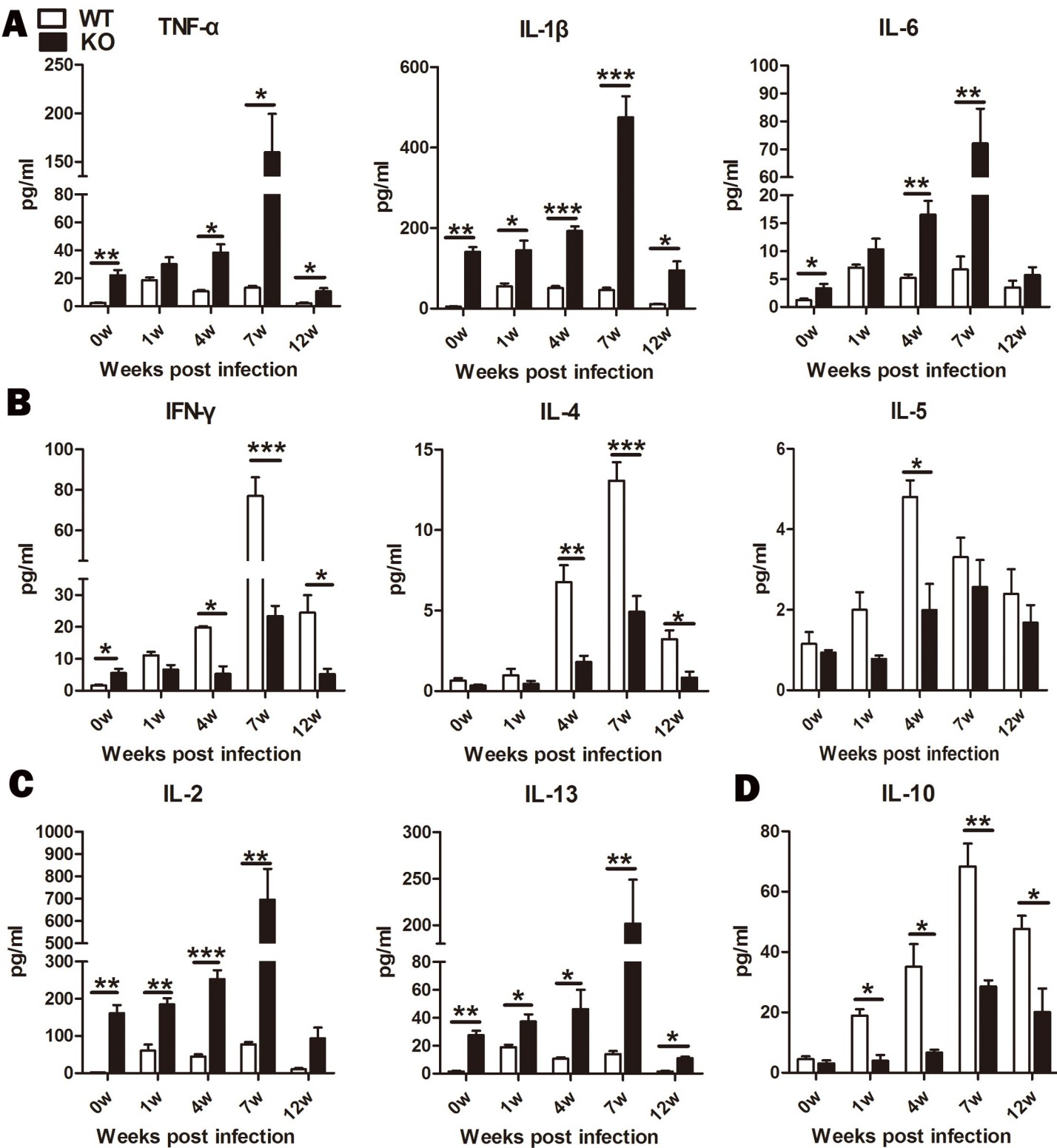

**Fig 3. A pro-inflammatory and pro-fibrogenic cytokine profile developed in infected iNOS-KO rats.** WT and KO rats were infected percutaneously with 200 *S. japonicum* cercariae and serum was obtained for cytokine expression analysis by Multiplex at 0, 1, 4, 7, and 12-weeks post-infection. (A) Pro-inflammatory cytokines TNF-α, IL-1β, and IL-6 were significantly elevated in infected KO rats. (B) Th1 and Th2 cytokines were down-regulated in infected KO rats. (C) IL-2 and IL-13 were significantly up-regulated in infected KO rats. (D) The immunosuppressive cytokine IL-10 was significantly down-regulated in infected KO rats. n = 5 rats per group. Data shown are mean ± SEM and repeated twice with similar results. Significant differences have been noted, *$P < 0.05$, **$P < 0.01$, ***$P < 0.001$.

### iNOS expression ensures maximal development of Th2 and Th1 responses in the rat during *S. japonicum* infection

Previous studies have reported that a Th2 response was strongly activated in the absence of available egg production in the rat model of schistosomiasis and speculated that the increased Th2 response was involved in the resistance to *Schistosoma* infection [4,9]. We also observed increased levels of the Th2-related cytokines IL-4 and IL-5 in serum from infected WT rats. Therefore, to determine whether the high levels of expression of iNOS were responsible for the elevated Th2 response in rats following infection with *S. japonicum*, Th1 and Th2 responses were compared between WT and iNOS-KO rats. Here, Th1 and Th2 responses were analyzed by the expression of IFN-γ and IL-4 in CD4$^+$ T cells from peripheral blood, LN, and spleen from 7-week *S. japonicum*-infected animals detected by flow cytometry. As shown in Fig 4A and 4B, no significant differences were observed in the frequency of CD4$^+$ T cells producing IL-4 in the blood, LN, and spleens between naïve WT rats and naïve KO rats ($P>0.05$). As expected, there was a marked increase in the frequency of IL-4-producing CD4$^+$T cells in the blood ($P<0.001$), LN ($P<0.01$), and spleens ($P<0.001$) of WT rats following infection with *S. japonicum*. However, in KO rats following infection with *S. japonicum*, the significant increase of IL-4-producing CD4$^+$ T cells frequency was only found in the blood but not in the LN and spleens, in which the magnitude of increase between naïve and infected groups was lower than those found in the WT rats. When the frequency of CD4$^+$ T cells producing IFN-γ was compared, similarly, there was a significant increase in the blood ($P<0.01$), LN ($P<0.01$), and spleens ($P<0.05$) of WT rats following infection with *S. japonicum* (Fig 4C and 4D). And the significant increases of IFN-γ-producing CD4$^+$ T cells frequency were also found in the LN and spleens of KO rats following infection with *S. japonicum*, but not in the blood (Fig 4C and 4D). No significant difference was observed in the blood, LN, and spleens between infected WT rats and infected KO rats ($P>0.05$) (Fig 4C and 4D). From these results, we cannot determine whether the high levels of expression of iNOS in rats is responsible for the elevated Th2 response during *S. japonicum* infection. Furthermore, the production of IFN-γ and IL-4 in splenocyte culture supernatants stimulated for 48 hours with schistosome worm antigens (SWA), was also detected. The results showed that the production of IFN-γ and IL-4 in splenocytes isolated from infected KO rats was significantly lower than that isolated from infected WT rats (Fig 4E and 4F), suggesting that both Th1 and Th2 responses are impaired in the absence of iNOS in rats during *S. japonicum* infection.

Our previous studies have shown that the adoptive transfer of wild-type macrophages to KO rats, which could elevate the production of NO *in vivo*, could partially recover the resistance to *S. japonicum* in KO rats [10]. Thus, to determine whether exogenous production of NO treatment could reverse the immune response profile of the KO rats, we separated the splenocytes from infected KO recipient rats (KO + Mφ) that received adoptive transfer macrophages and non-transferred controls and examined the production of IFN-γ and IL-4. As expected, when the adoptive transfer of wild-type macrophages was performed in KO rats, the production of IFN-γ and IL-4 in splenocytes was significantly increased compared with non-transferred controls (Fig 4G and 4H). Therefore, these results suggested that high expression levels of iNOS ensured maximal development of Th2 and Th1 responses in rats with schistosomiasis.

### Cytokine and chemokine production in local tissues (liver) is dependent on the expression of iNOS in rats during *S. japonicum* infection

The liver is the main location of pathological changes and schistosome parasitism in rats with schistosomiasis. To evaluate the role of iNOS in regulating the chemotaxis and effector effects

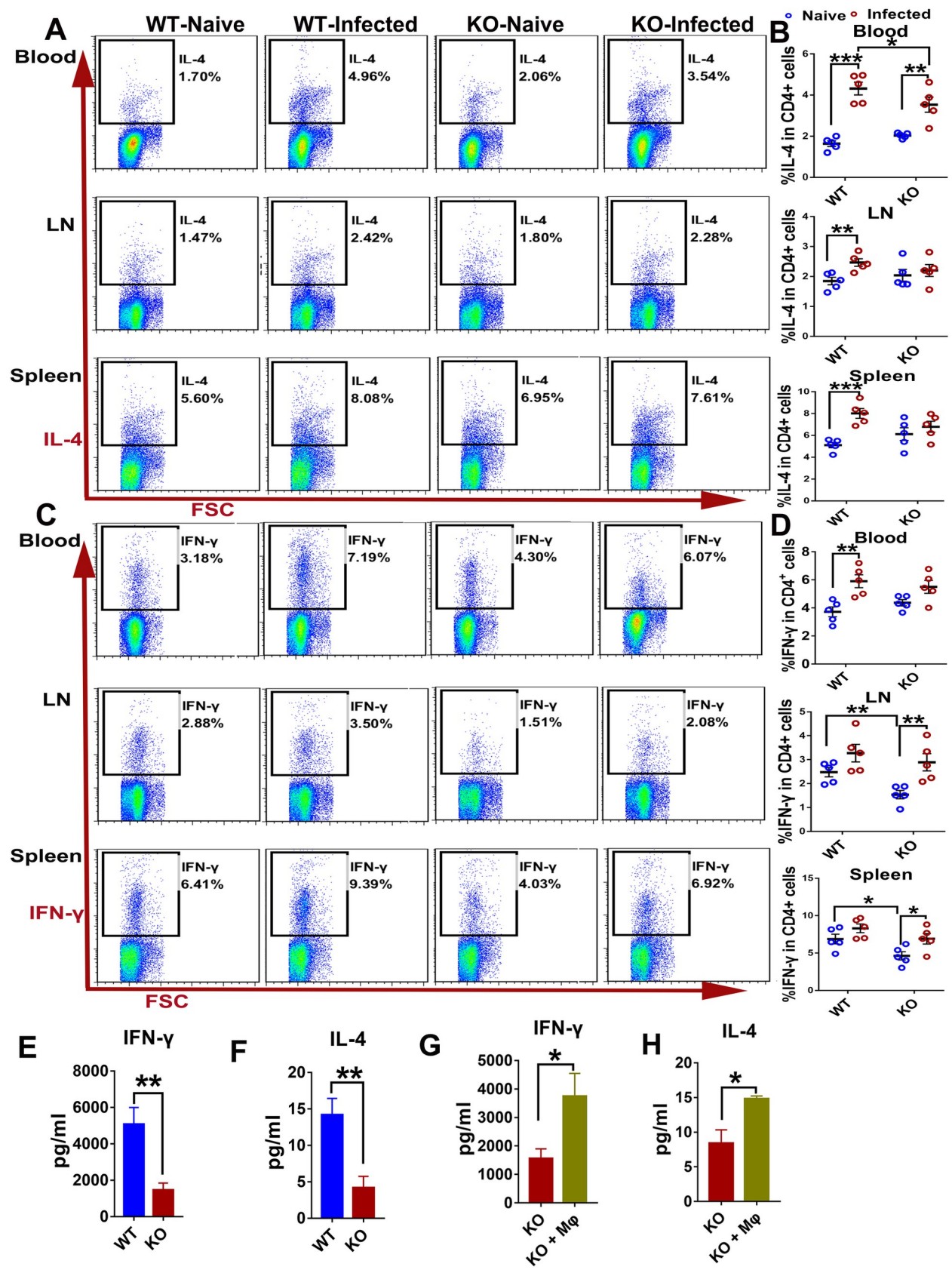

**Fig 4. iNOS deficiency impairs *S. japonicum*-elicited Th1 and Th2 responses in rats.** Peripheral blood, mesenteric Lymph Nodes (LN), and splenic leukocytes were isolated from naive and 7-week *S. japonicum*-infected WT and KO rats. Then they were stimulated with PMA/Ionomycin/ BFA for 6 hours for analysis of the production of IFN-γ and IL-4 in CD4+ T cells by flow cytometry. (A) Representative FACS plots of IL-4 producing CD4+ T cells in the blood, LN, and spleen. (B) Frequency of CD4+ T cells producing IL-4. (C) Representative FACS plots of IFN-γ producing CD4+ T cells in the blood, LN, and spleen. (D) Frequency of CD4+ T cells producing IFN-γ. Results for individual rats are shown and statistically significant differences are indicated. (E) (F) Splenic leukocytes from infected WT and KO rats were stimulated for 48 hours with SWA. IFN-γ and IL-4 concentrations in supernatants were measured. (G) (H) Adoptive transfer of wild-type macrophages was performed in KO rats as described in Materials and Methods (KO + Mφ), and the infected KO rats that only received a PBS injection were used as the control group (KO). The rats were sacrificed at 6 weeks post-infection. IFN-γ and IL-4 concentration in supernatants of splenic leukocytes collected from infected KO and KO + Mφ rats after stimulation for 48 hours with SWA were detected by ELISA. Data are expressed as the mean ± SEM of 4–5 rats per group. *P<0.05, **P<0.01,***P<0.001.

of immune cells involved in immunopathology and activity against *Schistosoma*, we compared the immunological characteristics of the livers of WT and KO rats. As shown in Fig 5A, local (liver) IFN-γ, IL-4, IL-10, TGF-β1 levels in infected KO rats were significantly lower than those in infected WT rats ($P<0.05$), which was consistent with the results found for serum (Fig 3). Surprisingly, the expression of IL-6 mRNA also showed a marked reduction in infected KO rats ($P<0.01$), in contrast to the serum which showed significantly increased expression (Fig 3A). Again, similar findings were confirmed by immunohistochemistry, which showed that the expression of IFN-γ, IL-10, TGF-β3, and IL-6 at the periphery of granulomas in KO rats were significantly lower than those in WT rats (Fig 5B and 5C). Furthermore, the expression of chemokines in the liver of infected KO rats was significantly decreased compared with that in WT rats, such as the monocyte chemoattractant protein (MCP)-1, Eotaxin, Eotaxin-2, neutrophil-activating protein (NAP)-3 (also named CXCL1), and macrophage inflammatory protein (MIP)-2a (also named CXCL2), indicating a decline in the ability to recruit immune cells to the inflammatory site (Fig 5A). The results demonstrate that both the cytokines and chemokines responsible for the capabilities of granuloma-associated lymphocytes were diminished in the absence of iNOS in rats. Similarly, the results were further confirmed by the rescue experiment using the adoptive transfer of WT macrophages to KO rats, which partially restored cytokine and chemokine expression at the periphery of granulomas in the liver of infected KO rats (Fig 5D-E). These provide evidence that the expression of iNOS is required to induce the production of cytokines and chemokines in the liver of rats during *S. japonicum* infection, which is, in turn, very important in the immune response against *Schistosoma*.

## iNOS also participates in promoting pulmonary granuloma associated fibrotic processes, but not inflammation, in wild type rats

To determine the possible immunoregulatory effects of iNOS in the formation and development of fibrosis in rats, we exploited the schistosome egg-induced pulmonary granuloma model [28]. Pulmonary granuloma-associated fibrosis was compared in the WT and iNOS-KO rats on days 7 and 14 post-challenge, following intravenous injection with 15,000 live eggs. We have described previously that there were no significant differences in granuloma size in both genotypes of animals on days 7 and 14 post-challenge, although the KO rats displayed a non-significant, slight decrease [10]. When pulmonary granuloma-associated fibrosis in WT and KO rats were compared by Masson's trichrome staining (Fig 6A and 6B), surprisingly, compared with WT rats, the KO rats showed an average of 78% increase in blue fibrosis staining on day 7 ($P<0.01$), the peak of the granulomatous response [29], while displaying a slight, non-significant, reduction (27%) in collagen deposition on day 14 post-challenge ($P>0.05$). The results showed that iNOS in rats inhibited the formation of fibrosis in the acute phase and promoted fibrosis in the chronic phase. Furthermore, in comparing the fibrotic process in KO and WT rats, WT rats displayed a slight increase in collagen deposition on day 14 compared

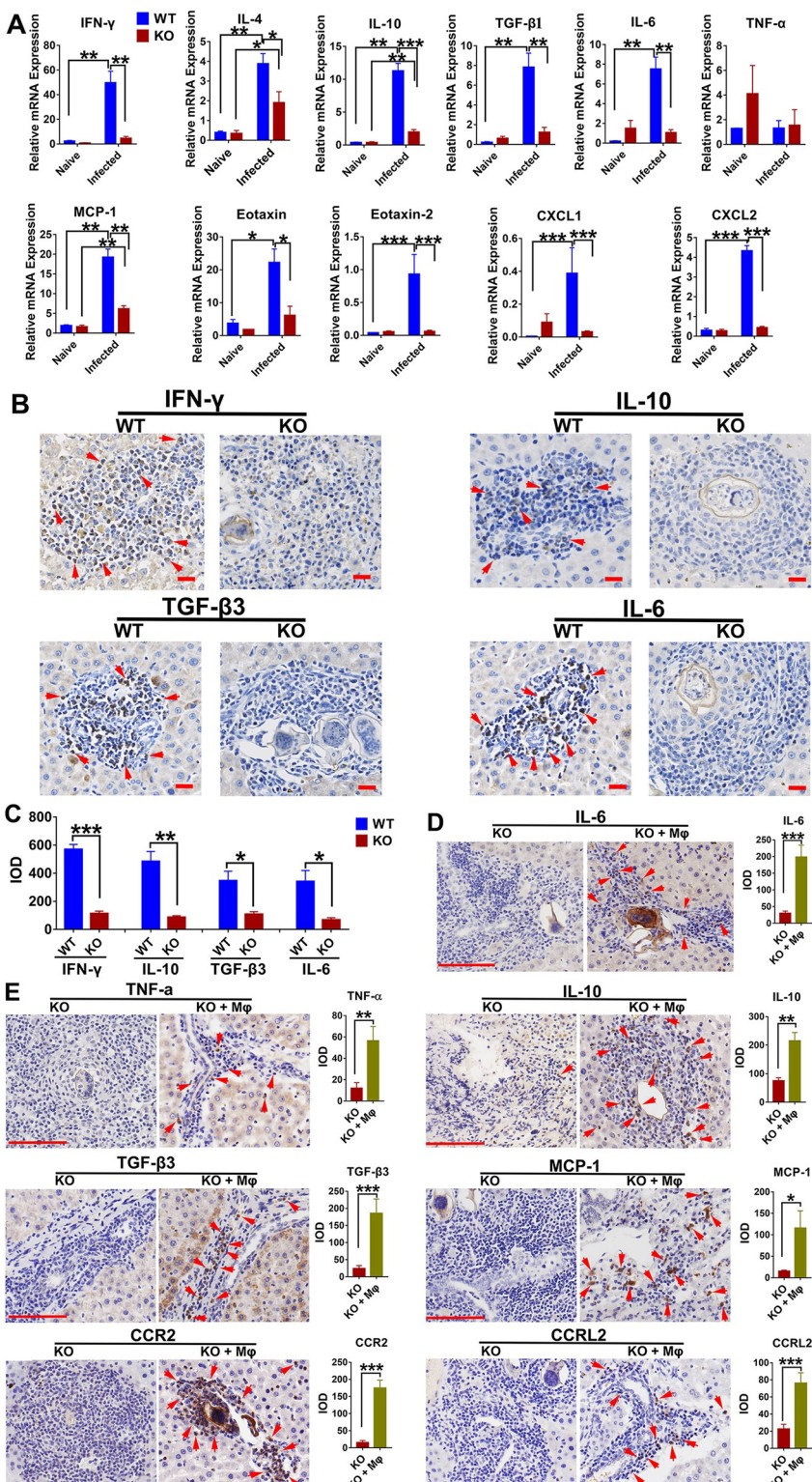

**Fig 5. Cytokine and chemokine production is impaired in the liver of infected iNOS-KO rats.** WT and KO rats were infected percutaneously with 200 *S. japonicum* cercariae and sacrificed at 7 weeks post-infection. (A) Relative mRNA levels of cytokines and chemokines in liver tissue were determined by real-time PCR. Expression is normalized to β-actin. Data shown are mean ± SEM and repeated twice with similar results, n = 5 rats per group. Significant differences have been noted, $^{*}P < 0.05$, $^{**}P < 0.01$, $^{***}P < 0.001$. (B) Hepatic granuloma-associated cytokine

expression was detected by immunohistochemistry (IHC). Brown (arrow) indicates a positive signal (Scale bars: 20 μm). (C) The statistical results of immunohistochemistry. IOD, integrated optical density. (D-E) Hepatic granuloma-associated cytokine and chemokine expression were detected by immunohistochemistry after adoptive transfer of macrophages from WT rats into iNOS-KO rats (KO + Mφ group). Scale bars: 100 μm. Brown (arrow) indicates a positive signal. The statistical results of the immunohistochemistry were shown as IOD. Significant differences have been noted, $^*P < 0.05$, $^{**}P < 0.01$, $^{***}P < 0.001$.

with day 7 post-challenge. In contrast, collagen deposition was markedly decreased in KO animals on day 14, with nearly 56% less than that found on day 7 post-challenge ($P<0.01$) (Fig 6B). In addition, fibrosis was also examined using immunohistochemistry with alpha-smooth muscle actin antibodies (α-SMA), a marker of activated myofibroblasts, as shown in Fig 6C

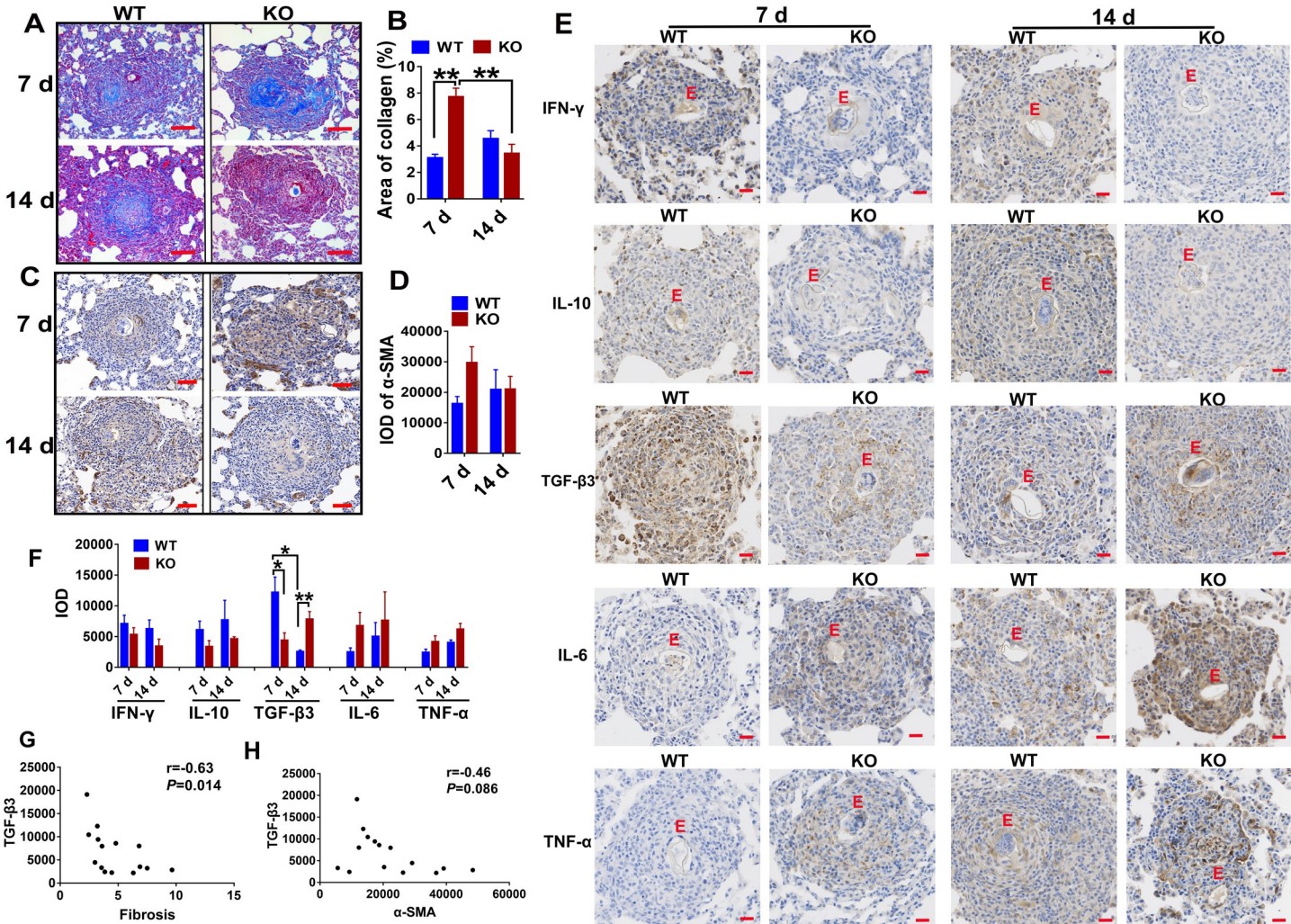

**Fig 6. The progression of pulmonary granuloma-associated fibrosis is slowed in iNOS-KO rats.** 15,000 eggs collected from *S. japonicum* infected rabbits were delivered to the lungs of WT and iNOS-KO rats via tail vein injection (i.v.), and lungs were removed for fibrosis analysis on 7 and 14 days post-challenge. (A) Representative images of pulmonary granuloma-associated fibrosis stained with Masson's Trichrome. Blue represents collagen fibers. (B) Fibrosis was quantified from Masson's Trichrome stained lung sections. (C) The immunohistochemistry results of α-SMA in lungs. Brown indicates the α-SMA signals. Scale bars = 50 μm. (D) Quantitation of the expression of α-SMA. (E) Representative images of pulmonary granuloma associated cytokines expression by immunohistochemistry. Brown indicates the positive signals (Scale bars: 20 μm). E, egg. (F) Quantitation of the expression of cytokines. The data are expressed as the mean ± SEM of 4 rats per group. Significant differences have been noted, $^* P<0.05$, $^{**} P<0.01$. IOD, integrated optical density. (G) Correlation between the level of TGF-β3 and fibrosis from Masson's Trichrome was analyzed by the Spearman rank correlation test (r = −0.63; P = 0.014). (H) Correlation between the level of TGF-β3 and α-SMA (r = −0.46; P = 0.086).

and 6D. Consistent with the collagen deposition by Masson's trichrome, KO rats also displayed much stronger staining for α-SMA on day 7 post-challenge and weakened on day 14 post-challenge (Fig 6C and 6D). The results showed that the fibrosis process is inhibited in KO rats at day 14 post-challenge, compared with WT rats which showed persistently elevated α-SMA expression. Therefore, these results suggest that iNOS accelerates the pulmonary granuloma-associated fibrotic process in rats.

To dissect the possible mechanisms for iNOS-mediated promoting pulmonary granuloma-associated fibrotic process in rats, we detected the expression of cytokines at the periphery of granulomas in WT and KO models by immunohistochemistry. As shown in Fig 6E and 6F, the KO rats displayed a slight, non-significant, reduction in the expression of IFN-γ and IL-10 and a slight, non-significant, increase in the expression of IL-6 and TNF-α on day 7 and 14 in comparison with WT rats ($P$>0.05). This is consistent with the results found in the serum of KO rats infected with *S. japonicum* (Fig 3). Interestingly, compared with WT rats, however, the expression of TGF-β3 around pulmonary granulomas in KO rats showed nearly a 64% reduction on day 7 ($P$<0.01) and 67% increase on day 14 post-challenge ($P$<0.05) (Fig 6E and 6F), which is opposite to the trend observed in pulmonary granuloma associated fibrosis. To further investigate whether the expression of TGF-β3 at the periphery of granulomas in lungs was associated with the development of fibrosis in rats, we conducted a correlation analysis between the expression level of TGF-β3 around pulmonary granuloma and pulmonary granuloma associated fibrosis in WT and KO rats. Fig 6G shows that the TGF-β3 levels were negatively correlated with fibrosis quantified using Masson's Trichrome in WT and KO rats (r = −0.63; $P$ = 0.014). No significant correlation was found between TGF-β3 levels and the expression of α-SMA at the periphery of granulomas (r = −0.46; $P$ = 0.086) (Fig 6H). The results indicate that TGF-β3 may be involved in the iNOS-mediated pathway promoting the pulmonary fibrotic process in rats.

## Discussion

The rat, a non-permissive host, develops immune responses directed at juvenile or adult *Schistosoma* that causes rapid elimination of all parasites and no typical egg-granuloma formation [4,5], while mice (permissive hosts) do not. Interestingly, schistosome eggs elicit a dominant Th2 immune response within mouse hosts [8,30], whereas rats with schistosomiasis develop significant a Th2 response in the absence of available egg production [4,9]. Our previous studies have demonstrated that the expression of iNOS and the levels of NO in peritoneal macrophages of rats are significantly higher than those in mice [10,14,15]. In the present study, our results showed that, following infection with *S. japonicum*, iNOS gene deficiency in rats led to slow clearance of parasites, increases in the development of worms, and an exacerbation of granuloma formation and fibrosis. Importantly, *S. japonicum*-elicited a Th2 and Th1 response that was significantly impaired in iNOS-KO rats. Therefore, this study demonstrates that the difference in iNOS levels between mice and rats may be responsible for the different immune responses and outcomes induced by schistosome infection in both hosts.

The results presented here show that the worm burden collected from the hepatic portal vein infusion of infected iNOS-KO rats was significantly increased compared with the infected WT rats at week 1 and 7 post-infection, while no significant differences were found at week 4 post-infection. Coulson, *et al.* [31] have concluded from their results that the effector response of NO, against an embolized parasite in the lungs, acts by blocking further migration rather than by direct cytotoxic killing. Therefore, in our study, the elevated worm load, at week 4, in WT rats may be attributed to the parasites that remained in the lungs having reached the liver. This is due to the migration of *S. japonicum* from the lung to the liver being inhibited by NO

in WT rats. This inference was further supported by the observations that the spines on the tegument in the middle part of the body of the schistosomula from iNOS-KO rats were much reduced compared to those found in WT rats (S1 Fig), that the loss of spines contributes to reducing the resistance during their migration [32]. Furthermore, NO in rats promotes the clearance of *S. japonicum* mainly after 4 weeks post-infection, such that the worm load decreased most obviously at 7 weeks after infection. The result implies that NO, as a cytotoxic agent, is more effective in eliminating the parasites in the livers than in the lungs, which may be due to the fact that NO can also function as a vascular relaxing factor to influence the ability of schistosomes to negotiate the vascular beds of the lungs so as to the migrating schistosomula can partially protect against the cytotoxic actions of NO. This opinion is supported by the results reported by Wilson *et al.* [31] that NO was not the major agent causing the pulmonary effector response to eliminate *S. mansoni* in mice exposed to the radiation-attenuated vaccine. It is worth noting that the iNOS-KO rats died at a significantly accelerated rate, with all animals succumbing within 5 weeks infected with a high dose of parasites, compared with WT rats. The significantly increased mortality may be attributed to the increased parasite burden and development-mediated inflammation, as demonstrated by the significant increase in pro-inflammatory cytokines (TNF-α, IL-6, and IL-1β) levels in the serum of infected iNOS-KO rats within 5 weeks post-infection.

Based on the results, we deduce that *S. japonicum*–infected iNOS-KO rats may be immuno-compromised and unable to clear the infection. Thus, we first examined whether the generation of immune cells was affected by iNOS deficiency. Our results showed that iNOS deficiency in rats led to a reduction in the frequencies of T cells and an increase in B cell numbers in mesenteric lymph nodes and livers, but had no effects on macrophages. These results suggest that the expression of iNOS may play an important role in maintaining the host T-cell immune response and inhibiting B cell populations in rats with schistosomiasis. T cells play an essential role in the immune response to schistosomiasis. It has been reported that the preferential proliferation of T cells in the draining lymph nodes of mice exposed to *S. mansoni* is central to the induction of protective immunity [33,34]. Furthermore, Lu *et al.* found that the absence of T cells in rats led to the loss of natural resistance to schistosomiasis [35]. In contrast, the role of B cells in the process of schistosome infection seems to be less important. Ferru *et al.* [36] demonstrated that B lymphocytes were not essential for the development of the general immune response towards *S. mansoni* in the mouse, they showed that the absence of B cells in mice did not affect the adult worm and egg burdens in comparison to control groups. Thus, it indicates that iNOS-mediated protective immunity against *S. japonicum* in rats may be mediated by maintaining T cell immune responses. CD4[+] and CD8[+] T cells are important for regulating the host's immune function against infection [37]. CD4[+] T cells participate in modulating the activity of the immune response mainly by regulating the secretion of cytokines and antibodies [38], and CD8[+] T cells take part in the cytotoxic effect [39]. In our study, a decrease in CD4[+] T cells was observed in naïve iNOS-KO rats compared to WT rats, which indicated a likely decline in immune response in iNOS-KO rats.

CD4[+] T cell-mediated Th1 and Th2 responses play an important role in against infection and immunopathogenesis of schistosomiasis. A strong Th1 response is highly desirable for effective host defense in the early stage of infection and is involved in inflammation, whereas Th2 response exerts anti-inflammatory effects and regulates Th1-mediated immunopathology [30,31,34,40]. Our results showed that compared with the infected WT rats, the frequency of IL-4-producing CD4[+]T cells was impaired in the spleen and the levels of Th2-associated cytokines (IL-4 and IL-5) in the serum and liver were remarkably reduced in the infected iNOS-KO rats. Surprisingly, the reduction in Th2 response did not result from an increased Th1 response, because the production of Th1-related cytokines in the serum, spleens, and liver

of infected iNOS-KO rats was also obviously decreased, in comparison to the infected WT rats. Previous studies reported that NO preferentially induced Th1 cell differentiation and Th1-related cytokines production by selectively up-regulating IL-12 receptor expression via cGMP [41]. This suggests that when NO is present in WT rats, NO may preferentially induce a Th1 response stimulated by schistosome antigens in the early stage of infection and participate in the anti-schistosome response. The continuously elevated Th1 response driven by NO, on the one hand, inhibits the survival of *Schistosoma*, and on the other hand, will initiate the host protective mechanism, that is, induce the increase of Th2 responses to inhibit the excessive Th1 pro-inflammatory response. However, in the absence of inducible NO in iNOS-KO rats, the Th1 response may not be effectively induced, and the host may not be able to start the Th2 response without an excessive Th1 response, which leads to the inability of *Schistosoma* to be killed. Therefore, these results indicate that iNOS expression ensures maximal development of Th2 and Th1 responses in the rat during *Schistosoma* infection. In addition, previous studies reported that the Th2 response plays an important role in resistance to *Schistosoma* infection in rats, that the worm burden was significantly increased in rats following treatment with anti-IL-4R or anti-IL-13R antibodies and which could be associated with their ability to control the production of IgE and eosinophils [4]. The downregulated Th1 and Th2 responses in iNOS-KO rats are not conducive to the elimination of parasites. Interestingly, these results found in infected iNOS-KO rats differ from the observations obtained from infected iNOS-KO mice, which develop significantly enhanced Th1-associated cytokine responses (IFN-γ) and lower expression of Th2-associated cytokines (IL-4, IL-5) levels in the lungs after vaccination with attenuated *S. mansoni* [42] and displayed partially reduced resistance [31,42]. Similar Th-associated cytokine response profiles were also found in the egg/IL-12-sensitized iNOS-KO mice, while there was little or no significant effect on the Th2 or Th1-type cytokine polarization during the natural course of infection with *S. mansoni* in the iNOS-KO mice *versus* WT mice, given the low expression of iNOS detected in infected mice [43]. These data indicate that iNOS activation is not a prerequisite essential mechanism for Th1 response development in mice. Thus, the results obtained with iNOS-deficient animals (rats and mice) accurately reflect that the difference in NO concentration in the host may lead to different levels of resistance to *S. japonicum* by regulating the immune response of the host.

Moreover, it is of note that the transcriptional levels of all cytokines and chemokines measured in the liver (the main location of schistosome parasitism in rats) of iNOS-KO rats were markedly lower than those of WT rats. Chemokines play an important role in regulating cell trafficking. When the parasites persist in a particular organ, the chemotactic signals facilitate the recruitment of macrophages, eosinophils, basophils, and T cells to form a granuloma to keep the parasites or eggs sequestered under control [44]. Therefore, the down-regulated chemokines in iNOS-KO rats are not conducive to the recruitment of inflammatory cells to the periphery of worms and therefore were unable to secrete abundant cytokines necessary for rapid elimination of *Schistosoma*. These results demonstrate that both the cytokine and chemokine-producing capabilities in the liver are linked to the expression of iNOS in rats. This notion is consistent with the previous findings in iNOS-KO mice that the size of inflammatory foci around schistosomula in the lungs was diminished [42]. It is worth noting that the expression of IL-6 in serum is inconsistent with that in the liver. IL-6 is also an inflammatory cytokine and it was elevated in the serum of iNOS-KO rats. It is considered that this might be derived from several different damaged organs, besides the liver, reflecting the overall level of inflammation in the host, and this might be attributed to the significantly increased parasite burden and development-mediated inflammation. In addition, the developed proinflammatory immune environment in naïve iNOS-deficient rats, with an increase in TNF-α, IL-1β, and IL-6 expression compared with naïve WT rats, may be more conducive to the development of

schistosomes. Previous studies have demonstrated that pro-inflammatory stimuli could promote schistosome development and exacerbate infection [44]. A similar observation was also demonstrated by Riner *et al* [45], who revealed that repeated administration of an endogenous inflammatory stimulus could restore schistosome development in recombination activating gene-deficient (RAG$^{-/-}$) mice, which fail to induce liver inflammation and necrosis after infection with *S. mansoni*. In addition, some studies showed that TNF could indirectly stimulate schistosome development by binding to the TNF receptor and TLR signalling pathway [46–49]. Thus, based on our findings, we demonstrate that the high levels of expression of NO in rats can inhibit the development of *S. japonicum* by regulating the host immune system.

The role of iNOS in fibrosis is highly controversial, with some studies showing pathogenic [45–47] and protective [43,48] roles for this enzyme. Notably, in our present study, the increased fibrosis in the liver of infected iNOS-KO rats may be mainly due to the marked increased egg production and the viability of parasite eggs [10], which formed larger-sized granulomas and developed into more severe fibrosis in comparison to infected WT rats. Therefore, we could not determine the protective effect of NO on fibrosis in rats. With the schistosome egg-induced pulmonary granuloma model, interestingly, we found that iNOS played different roles during acute versus chronic stages. Pulmonary granuloma-associated fibrosis was inhibited at the acute stage (7 days post-challenge) by the expression of iNOS in rats, while such protection was completely abolished during the chronic stage (14 days post-challenge). In other words, these results suggest that the expression of iNOS accelerates the pulmonary granuloma-associated fibrotic process in rats. These observations were consistent with the results reported in allergen exposure-induced pulmonary fibrosis models that fibrosis was completely absent in iNOS-KO mice when using the chronic protocol [49]. The reason for this result may be that the iNOS gene deletion caused a failure to induce the infiltration of fibroblasts and collagen production in the chronic stage. While the presence of iNOS promoted the development of fibrosis, induced by inflammation, and suggests a possible cause of rapid inflammatory repair in the lungs and liver of rats with schistosomiasis. Studies in a wide range of experimental models have demonstrated the profibrotic actions of TGF-β1 [50–53]. In contrast, some experimental studies indicated that TGF-β2 and TGF-β3 can exert antifibrotic effects [54–56]. In the present study, the result showed that the effect of iNOS on egg-induced lung fibrosis was negatively correlated with the expression of TGF-β3, which suggests that TGF-β3 may protect against the development of pulmonary fibrosis. This idea confirms earlier studies showing that TGF-β3 decelerates the progress of radiation-induced pulmonary fibrosis by hindering fibrocyte recruitment and regulating the IFN-γ/IL-4 balance [55].

In conclusion, with the iNOS-KO rat model, the results reveal that iNOS plays a critical role in the regulation of immune system reactivity, particularly in Th1/Th2-associated cytokine responses and chemokine production in rats, during *S. japonicum* infection, which leads to a rapid rejection of schistosome survival and no granuloma formation. Furthermore, we provide direct evidence that high levels of NO in rats can promote the development of fibrosis, induced by inflammation, and suggest a possible cause of rapid inflammatory repair in the lungs and liver of rats with schistosomiasis. The key interactions, reported in this study, are schematically represented in Fig 7. Thus, this study significantly enhances our understanding of the immunoregulatory effect of NO on defense and immunopathological responses in rats. However, using the systemic knockout iNOS gene rat model, rather than a tissue-specific or cell-specific conditional knockout model, we cannot determine which iNOS-expressing tissues or cell subsets are the primary drivers of the *S. japonicum* susceptibility phenotype.

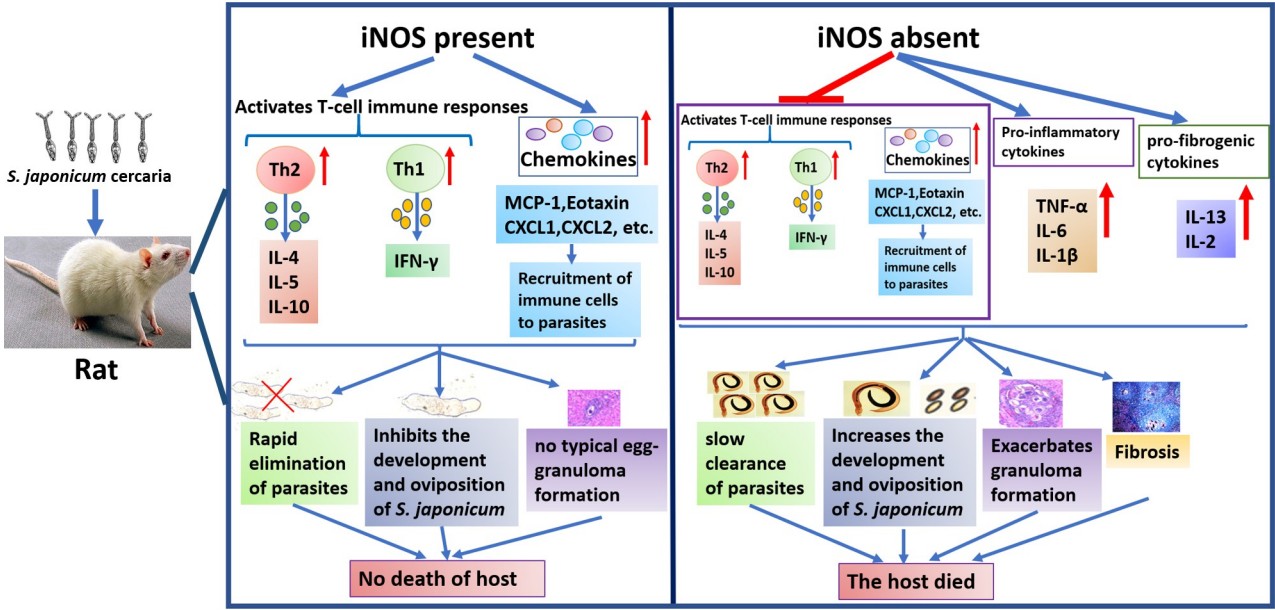

**Fig 7. Schematic model for the role of iNOS in maintaining host immune activity against *S. japonicum* infection in rats.** When iNOS is present in rats, T-cell immune responses were rapidly activated upon infection with *S. japonicum*, with associated consequences: Th2 and Th1 immune cells upregulated, and the secretion of cytokines and chemokines increased. These immune effects mediated by iNOS lead to rapid elimination of schistosome, inhibition of worm development, and no typical egg-granuloma formation in rats. Therefore, schistosomiasis does not cause death in rats. However, T cell responses were impaired in the absence of iNOS in rats, which resulted in delayed clearance of worms, increasing the development and oviposition of schistosomes, a lot of egg-granuloma formation which developed into fibrosis and produced a lot of inflammatory cytokines, eventually leading to the death of the host.

## Materials and methods

### Ethics statement

All animal work was approved by the Laboratory Animal Use and Care Committee of Sun Yat-Sen University under license number 2012CB53000 and conformed to the Chinese National Institute of Health Guide for the Care and Use of Laboratory Animals.

### Animals and parasite infections

Sprague Dawley (SD) rats were purchased from the Medical Experimental Animal Center of Guangdong Province. The iNOS-KO rats, SD strain purchased from the Beijing Vital River Laboratory Animal Technology Co., Ltd. (Beijing, China), were generated with the transcription activator-like effector nuclease technology (TALENs) [57]. These deficient rats are viable, fertile, and do not display any obvious appearance or physical abnormalities. Animals were kept in ventilated cages with available food and water under specific-pathogen-free conditions in the Sun Yat-Sen University following the university policy. Six to eight-week-old male rats were used for the study. In the infection experiments, rats and rabbits were percutaneously infected with 200 or 1000 cercariae of a Chinese mainland strain of *S. japonicum* obtained from infected *Oncomelania hupensis* snails, which were purchased from the Shanghai Institute of Parasitic Diseases (Shanghai, China). Parasites were perfused from the portal tract at 1, 4- and 7-weeks post-infection and kept in 0.9% saline. The collected worms were counted and photographed under a stereoscopic microscope (M205FA, Leica, Germany). To obtain worms for antigen preparation (schistosome worm antigens, SWA), *S. japonicum* was collected by perfusion of the hepatic portal system on day 28 post-infection from WT rats. After being

washed three times in PBS, the worms were homogenized in normal saline and subjected to 5 cycles of rapid freezing and thawing. The suspension was centrifuged at 12 000 rpm for 30 min at 4˚C, and the supernatants were collected and stored at −80˚C. Protein concentrations were determined by a BCA kit. For the induction of pulmonary granulomas, *S. japonicum* eggs were isolated and purified from the livers of infected rabbits. 15,000 eggs were delivered to the lungs of WT and iNOS-KO rats via tail vein injection (i.v.) [10,28]. Animals were sacrificed on days 7 and 14 post-challenge, and the left lung was removed for fibrosis analysis.

## Histopathology and fibrosis

Pulmonary and liver tissues were fixed in 4% neutral buffered formalin, embedded in paraffin for sectioning, dewaxed, and stained with Masson's trichrome for fibrosis analysis. Collagen is shown as blue in Masson staining, and the blue area reflects the amount of collagen. The severity of fibrosis was determined by the area of collagen using Image-Pro Plus 6.0 software. All sections were imaged using an automatic slide scanning system (AxioScan.Z1, Zeiss, Germany) at 10X magnification.

## Immunohistochemistry

For immunohistochemical analysis, the dewaxed sections were boiled in 10 mmol/L citrate buffer for 20 min for epitope retrieval following washing three times in PBS. After incubating in 3% hydrogen peroxide, the sections were blocked with 1% BSA for 1 hour. The slices were incubated with anti-rat antibodies against IL-10 (1:200, Servicebio, China), TGF-β3 (1:200, Servicebio, China), IL-6 (1:400, Servicebio, China), TNF-α (1:300, Servicebio, China), IFN-γ (1:150, Servicebio, China), α-SMA (1:1000, Servicebio, China), MCP-1 (1:500, Servicebio, China), CCR2 (1:800, Servicebio, China) and CCRL2 (1:200, Servicebio, China) at 4˚C overnight and then incubated with HRP labeled Goat anti-rabbit IgG general secondary antibody (DAKO, Denmark). Finally, the antibody binding was visualized by using a DAB kit (Thermo Scientific). All sections were imaged using an automatic slide scanning system (AxioScan.Z1, Zeiss, Germany) at 20X magnification.

## Flow cytometry

Spleens, mesenteric lymph node (LN), and livers from WT and iNOS-KO rats were removed aseptically and dispersed through a 70-μm nylon strainer. Single-cell suspensions of peripheral blood, spleens, LN, and livers were prepared by Ficoll-Hypaque (Hao Yang Biological Manufacture, Tianjin, China) gradient centrifugation, washed twice, and suspended in PBS containing 0.1% BSA and 0.05% sodium azide. For surface staining, $2 \times 10^6$ cells per 100 μl were incubated with CD3-APC (eBioscience, CA), CD19-FITC (Abcam, UK), CD4-PerCP-eFluor710 (eBioscience, CA), CD8-PE-Cy7 (eBioscience, CA), and macrophage marker-PE (eBioscience, CA) for 30 min at 4˚C in the dark. For the detection of intracellular cytokines, cells in complete RPMI 1640 medium were stimulated with PMA (20 ng/ml), Ionomycin (1 μg/ml), and Brefeldin A (10 μg/ml) (Sigma-Aldrich, Germany) for 6 h. Surface molecule staining CD3-FITC (BD Biosciences, USA), CD4-PE-Cy7 (eBioscience, CA), and CD8-PerCP (eBioscience, CA) for 30 min at 4˚C was carried out. Cells were fixed in 0.4% paraformaldehyde for 20 min, permeabilized with 0.1% saponin buffer (Sigma-Aldrich, Germany), and stained for IFN-γ-APC (BD Biosciences, USA) and IL-4-PE (BD Biosciences, USA). The expression of surface molecules and intracellular molecules were analyzed on a BD LSR II flow cytometer using FlowJo v.8 software (Treestar, CA).

## Cytokine analysis

The cytokines in serum samples were assayed using a rat 14 plex cytokine kit (eBioscience, CA) according to the manufacturer's instructions. Samples were read using a Bio-Plex Suspension Array System (Biorad, USA). The concentration range of protein standard in the standard curve of TNF-α, IL-1β, IL-6, IFN-γ, IL-4, IL-5, IL-2, IL-13, IL-10 was 1.44–11800 pg/ml, 6.5–53100 pg/ml, 1.09–8950 pg/ml, 1.67–13700 pg/ml, 0.31–2550 pg/ml, 0.71–5800 pg/ml, 0.91–7450 pg/ml, 1.5–3075 pg/ml, 3–24600 pg/ml, respectively, which covered all the data we obtained in this study. For determining the concentrations of IFN-γ and IL-4 in splenocytes culture supernatants, purified single-cells of spleen were plated in 96-well culture plates at a final concentration of $2 \times 10^5$ cells/ml in complete RPMI 1640 medium and stimulated with SWA (20 μg/ml) in triplicate and incubated at 37˚C in a humidified atmosphere of 5% $CO_2$ for 48 h. The concentrations of cytokines IFN-γ, IL-4 in harvested culture supernatants were detected by ELISA following the manufacturer's instructions (BD Bioscience, USA). All the data is within the range of quantification, when the levels of cytokines were over the max detection, the samples were diluted and re-detected.

## RNA isolation and real-time PCR

Liver tissues were placed in a 500 μl Trizol reagent (Invitrogen, USA) and mashed with Tissue-Lyser II (Qiagen, USA). Total RNA was isolated and further purified using RNeasy Mini Kit (Qiagen, USA) following the manufacturer's instructions. The concentration of RNA was quantified using a NanoDrop ND-1000 spectrophotometer (Nanodrop, USA). First-strand cDNA was synthesized using isolated RNA, Superscript II reverse transcriptase (Invitrogen, USA), and oligo dT as a primer. Genes expression was analyzed by quantitative real-time PCR (qRT-PCR) using the LightCycler480 real-time PCR system (Roche, Switzerland) and SYBR green qPCR Master Mixes (Roche, USA). The relative expression of each gene was normalized to the mean values for β-Actin before statistical analysis. β-Actin primers were: forward 5'-TGGAATCCTGTGGCATCCATGAAAC-3', reverse 5'- TAAAACGCAGCTCAGTAACA GTCCG-3'. IL-4 primers were: forward 5'- CACCCTGTTCTGCTTTCTCA-3', reverse 5'-CTCAGAGGGCTGTCGTTACA-3'. IFN-γ primers were: forward 5'- AGGCCATCAGCAAC AACATA-3', reverse 5'- AGCTTTGTGCTGGATCTGTG-3'. IL-10 primers were: forward 5'-CCCAGAAATCAAGGAGCATT-3', reverse 5'- TCATTCTTCACCTGCTCCAC-3'. TGF-β1 primers were: forward 5'- CTTGCCCTCTACAACCAACA-3', reverse 5'- CTTGCGACCCA CGTAGTAGA-3'. IL-6 primers were: forward 5'- ACTTCACAAGTCGGAGGCTT-3', reverse 5'- AGTGCATCATCGCTGTTCAT-3'. TNF-α primers were: forward 5'- TAGCAAACCACC AAGCAGAG-3', reverse 5'- CCACCAGTTGGTTGTCTTTG-3'. MCP-1 primers were: forward 5'- AATGAGTCGGCTGGAGAACT-3', reverse 5'- CTGGACCCATTCCTTATTGG-3'. Eotaxin primers were: forward 5'- CAGCTCTCCACAGCACTTCT-3', reverse 5'- GTCAT GGTAAAGCAGCAGGA-3'. Eotaxin-2 primers were: forward 5'- CTCCACCACCATCA TTGCTA -3', reverse 5'- GAAATAAAGGTCACGCAGCA -3'. CXCL1 primers were: forward 5'- CACCCAAACCGAAGTCATAG-3', reverse 5'-TTCAGGGTCAAGGCAAGC-3'. CXCL2 primers were: forward 5'- CGCCCAGACAGAAGTCATAG -3', reverse 5'- GACGATCCTCT GAACCAAGG -3'. Relative expression was calculated using the $2^{-\Delta\Delta CT}$ method.

## Scanning electron microscopy

Parasites collected from infected mice, WT, and iNOS-KO rats at 7-day post-infection were fixed individually with 0.2 M PBS containing 2.5% glutaraldehyde (pH 7.4) at 4˚C for 24 h. After washing three times with PBS and six times with distilled water, the samples were dehydrated in gradient ethanol and ethanol was exchanged with acetone and isoamyl acetate. The

tegument of the schistosomula was observed and photographed using a scanning electron microscope (S-2500, Hitachi) following critical point-drying and being coated with gold.

### Adoptive transfer experiments

In this work, the adoptive transfer of macrophages from WT rats into iNOS-KO rats was performed as previously described [10]. Briefly, after harvesting the peritoneal macrophages from WT rats, $1 \times 10^8$ cells suspended in PBS were transferred into iNOS-KO rats through the tail vein on days 0, 7, 14, 21, 28, and 35 post-infection with *S. japonicum* (KO + Mφ), and the KO rats that received only PBS were used as the control group (KO). The rats were killed at 6 weeks post-infection to investigate the production of cytokines and chemokines in splenic leukocytes and the liver.

### Statistical analysis

All statistical analyses were performed using SPSS 19 software. Significant differences between the two groups were determined using a student's unpaired T-test with Welch's correction or one-way ANOVA or the F-test. Spearman rank correlation test was used for correlation analysis. Graphs and analyses were performed using GraphPad Prism version 7.00 for Windows, San Diego, CA, USA. All data shown are presented as the mean ± SEM, and *P* values ≤0.05 were considered statistically significant.

### Supporting information

**S1 Data. Excel spreadsheet containing, in separate sheets, the underlying numerical data and statistical analysis for Figs panels** 1A, 1D and 1E, 2B and 2D, 3A, 3B, 3C and 3D, 4B, 4D, 4E, 4F, 4G and 4H, 5A, 5C, 5D and 5E, 6B, 6D, 6F, 6G and 6H.
(XLSX)

**S1 Fig. Scanning electron microscopy (SEM) analysis of the tegument of schistosomula from BALB/c mouse, WT rat, and iNOS-KO rat at 7 days post-infection. vs, ventral sucker; S, spine. Bar = 20 µm.** The loss of spines in the middle part of the body of schistosomula was found in mice and iNOS-KO rats, while large numbers of spines on the tegument are still present in the middle part of the body of schistosomula obtained from WT rats, suggesting parasite growth retardation.
(TIF)

**S2 Fig. Flow cytometry analysis of the changes in macrophage proportions in iNOS-KO and WT control rats.** WT and KO rats were infected percutaneously with 200 *S. japonicum* cercariae and sacrificed at 7 weeks post-infection. (A) Representative FACS plots of macrophages in the spleens, LN, and livers. (B) Frequency of macrophages in the spleens, LN, and livers. Results for individual rats are shown and statistically significant differences are indicated. Data shown are mean ± SEM and repeated twice with similar results. n = 5 rats per group. No significant differences were found. LN = lymph node.
(TIF)

### Acknowledgments

We would like to thank Juan Shen for assistance with the flow cytometry experiments and Yaqiong Wang for scanning electron microscope (SEM) imaging.

## Author Contributions

**Conceptualization:** Jia Shen, Zhong-Dao Wu, Zhao-Rong Lun.

**Data curation:** Jia Shen, Si-fei Yu, Mei Peng.

**Formal analysis:** Jia Shen, Mei Peng.

**Funding acquisition:** Jia Shen, Zhong-Dao Wu.

**Investigation:** Jia Shen, Zhao-Rong Lun.

**Methodology:** Jia Shen, Si-fei Yu, Mei Peng, De-Hua Lai, Geoff Hide.

**Project administration:** Zhong-Dao Wu, Zhao-Rong Lun.

**Resources:** De-Hua Lai, Zhao-Rong Lun.

**Supervision:** Zhao-Rong Lun.

**Writing – original draft:** Jia Shen.

**Writing – review & editing:** Jia Shen, Si-fei Yu, Mei Peng, De-Hua Lai, Geoff Hide, Zhong-Dao Wu, Zhao-Rong Lun.

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
