## [Decision Letter · Decision Letter 0]

29 Sep 2021

Dear Dr. Zhao-Rong Lun

Thank you very much for submitting your manuscript "iNOS is essential to maintain host immune activity against Schistosoma japonicum infection in rats" for consideration at PLOS Neglected Tropical Diseases. As with all papers reviewed by the journal, your manuscript was reviewed by members of the editorial board and by several independent reviewers. In light of the reviews (below this email), we would like to invite the resubmission of a significantly-revised version that takes into account the reviewers' comments. 

We cannot make any decision about publication until we have seen the revised manuscript and your response to the reviewers' comments. Your revised manuscript is also likely to be sent to reviewers for further evaluation.

Sincerely,

Alessandra Morassutti, PhD

Associate Editor

Walderez Dutra

Deputy Editor

Reviewer's Responses to Questions

**Key Review Criteria Required for Acceptance?**

**Methods**

-Are the objectives of the study clearly articulated with a clear testable hypothesis stated?

-Is the study design appropriate to address the stated objectives?

-Is the population clearly described and appropriate for the hypothesis being tested?

-Is the sample size sufficient to ensure adequate power to address the hypothesis being tested?

-Were correct statistical analysis used to support conclusions?

-Are there concerns about ethical or regulatory requirements being met?

Reviewer #1: 1.Can the authors specify the genetic background of iNOS-KO rats?

2.Please provide the Image-Pro Plus software version.

3.Please provide the limit of quantitation of each assay and check whether all the data is within the range of quantification.

4.Spacing errors in units (500 μL, 0.2 M, and so on). Please check it carefully.

5.In the section "RNA isolation and Real-time PCR", IFN-γ primers are duplicated, please check it carefully.

Reviewer #2: MAJOR CRITIQUES

The authors demonstrate that the iNOS-KO rat is a powerful model to investigate immunity to Schistosoma japonicum and shows no abnormal phenotype at baseline (pg 21). However in the absence of orthogonal lines of evidence, it is difficult to determine whether the experimental results are truly due to the direct involvement of iNOS and NO signaling—which is the authors’ central claim—or are confounded by secondary downstream effects of the iNOS knockout (such as abnormal early rat development). To provide the necessary evidence to support the central claim of their paper, the authors would need to perform the following new investigations:

- Rescue the iNOS knockout phenotype (partially or fully), via genetic re-introduction of an iNOS transgene into the knockout animals and/or via adoptive transfer of leukocytes (and particularly T cells) from WT rats into iNOS-KO rats.

- Use a non-genetic method, such as a small molecule iNOS inhibitor, to disrupt iNOS signaling in WT rats and demonstrate similar effects on S. japonicum and anti-parasite immune responses.

MINOR CRITIQUE

For Fig 6, panels G and H, the authors should consider whether a Spearman rank correlation would be more appropriate for analysis, rather than the Pearson correlation. It is unclear that TGF-beta 3 would be expected to have a parametric (linear) correlation with either fibrosis or alpha-SMA. If the relationship is non-linear, Spearman rank would be more appropriate and more sensitive to detect a correlation. 

The other experimental methods presented are appropriate and executed correctly. The sample sizes were appropriate. There are no ethical or regulatory concerns.

**Results**

-Does the analysis presented match the analysis plan?

-Are the results clearly and completely presented?

-Are the figures (Tables, Images) of sufficient quality for clarity?

Reviewer #1: 1.Authors wrote ‘When infected with a low dose of parasites, no death was found both in WT or KO rats...', what are the bases for choosing the low or high doses of parasites？

2.Authors wrote "This suggested an attenuation of T cell function and increased B cell function...".Actually, the author only showed the frequency of T cells and B cells.

3.Authors wrote "To evaluate the role of iNOS in regulating the host’s T cell immune function...". In fact, besides αβ T lymphocytes（CD4 and CD8 T cells), γδ T lymphocytes are unconventional immune cells, which have both innate- and

adaptive-like features allowing them to respond to a wide spectrum of pathogens. I think the results of the frequency and function of γδ T lymphocytes in rats during Schistosoma japonicum infection should be provided.

4.Please provide the statistics results of immunohistochemistry.

Reviewer #2: MINOR CRITIQUES

The scanning EM results in Fig S2 are only presented in Pg 15 of the Discussion section, however the findings from these studies should be presented in the Results section, probably in the section of text discussing Fig 1 (pg 7). 

Fig 5A: The bracket annotations showing significant p-values comparing different bars on the bar graph are missing for Eotaxin-2 and CXCL1.

**Conclusions**

-Are the conclusions supported by the data presented?

-Are the limitations of analysis clearly described?

-Do the authors discuss how these data can be helpful to advance our understanding of the topic under study?

-Is public health relevance addressed?

Reviewer #1: I have no comments.

Reviewer #2: A major weakness throughout this paper is that, at several points, the authors report causal links between iNOS, immune cell subsets, and protection against parasite infection, however these conclusions are not supported by the data presented and the limitations of their analyses are not adequately described. There are multiple examples where the conclusions overstep the data and/or the effect of the iNOS knockout is described too vaguely:

Pg 9: “Thus, the results reveal that NO is necessary for maintaining the host’s cellular immune defense against infection in wild-type rats.” 

Pg 17: “NO is necessary for maintaining the host’s cellular immune defense against S. japonicum in wild-type rats.”

Fig 2 only shows that iNOS-KO rats have different frequencies of B cells, T cells, and T cell subsets during infection, but this does not equate to NO being necessary for “cellular immune defense against infection.” Furthermore, in Panel E, there are no significant differences between infected WT and infected KO rats in CD4/CD8 ratio, so there is no evidence for a relationship between CD4/CD8 ratio and protection against schistosome infection. To demonstrate that iNOS is necessary for cellular immunity, as claimed, the knockout phenotype needs to be rescued experimentally by adoptive transfer, as outlined in my Methods critique above. 

Pg 11: “The results suggest that high expression levels of iNOS are required to sustain CD4+ T cell responses in rats with schistosomiasis.” This claim is not supported by the preceding data in Fig 4. In panels B and D, in infected rats there are no significant differences between WT and KO in % IL-4 positive or % IFN-g positive CD4 T cells. Thus, the absence of iNOS had no effect on the frequency of IL-4 or IFN-g-producing CD4 T cells during schistosome infection. 

Pgs 11-12: The authors report the strikingly discordant results in IL-6 expression between the serum and the liver, however they do not comment on why this discordance exists and its potential implications of immunity to S. japonicum.

Pg 13: The authors use the results from the schistosome egg-induced pulmonary granuloma model to claim that “the exacerbated granuloma-associated inflammation and fibrosis in the liver of iNOS-KO rats were attributed to the increased worm burden, egg load, and viability, but not the effects of iNOS on pathology.” This claim assumes that the mechanisms that mediate fibrosis in the lung and the liver are similar, however this is not the case. The mechanisms of fibrosis are quite tissue-specific, and cannot be analogized between different tissues. For instance, cirrhosis is a fibrotic mechanism that is unique to the liver, and does not occur in any other tissue. Thus, the effects of iNOS on worm burden, egg load, and viability may be due to liver-specific effects that cannot be recapitulated in a pulmonary granuloma model, so lung fibrosis data cannot be extrapolated to the liver context.

Pg 15: “S. japonicum-elicited a Th2 response that was significantly impaired in iNOS-KO rats. Therefore, this study demonstrates that the difference in iNOS levels between mice and rats is responsible for the different immune responses and outcomes induced by schistosome infection in both hosts.” This is an incomplete interpretation of the data, as Th1 responses were also significantly impaired—specifically IFN-g—both systemically (Fig 3B) and in the liver (Fig 5A). Indeed, the authors need to expand on their interpretation of why iNOS-KO leads to suppression of both Th1 and Th2 cytokines, and to be more precise in how this affects existing models of the Th1/Th2 balance in anti-schistosomal immunity. Furthermore, there is no data in this paper comparing immunity and infection outcomes between mice and rats, so the second claim is unsupported.

Pg 15: The authors interpret the data from Fig 1A as implying “that it is in the liver rather than in the lungs that NO is effective in eliminating the parasite.” However, the authors do not suggest any potential mechanism to explain the early parasite clearance seen at the 1 week time point (albeit of a smaller magnitude than the clearance seen at 7 weeks). 

Pg 19: “...the expression of iNOS accelerates the pulmonary granuloma-associated fibrotic process in rats.” This interpretation contradicts the data in Fig 6B, which suggests that iNOS actually impairs early fibrosis (7 days) but shows no significant difference between WT and KO by 14 days.

The authors do not discuss a major limitation of the the iNOS-KO model—it is unclear which iNOS-expressing tissues or cell subsets are the primary drivers of the S. japonicum susceptibility phenotype. This could be addressed using tissue-specific and/or inducible knockout systems, but at a minimum, this limitation should be stated in the discussion. 

Finally, the Discussion needs to focus more on potential mechanisms by which iNOS exerts its effects. Upon what molecules is NO acting in this context? What are the signaling pathways activated by NO that may lead to the observed immune phenotypes?

**Editorial and Data Presentation Modifications?**

Reviewer #1: The authors report the immunomodulating activities of iNOS in rats, during S. japonicum infection, which leads to a rapid rejection of schistosome survival and no granuloma formation. Some interesting data were obtained in tests of the immuno assays and anti-granuloma formation activity.

However, upon initial review, several aspects are lacking and require revision before it can be considered further.

Reviewer #2: There are multiple grammatical and typographical errors that need to be reviewed and corrected.

**Summary and General Comments**

Reviewer #1: (No Response)

Reviewer #2: This manuscript by Shen et al. describes an iNOS knockout rat model of Schistosoma japonicum infection. The authors demonstrate that iNOS deficiency enhances S. japonicum parasite burden and disease pathology and describe differences in lymphocyte subsets as well as cytokine and chemokine expression associated with iNOS knockout. Strengths of the study include the detailed characterization of the immune and tissue phenotypes of the knockout rat, particularly when the rats are infected. Novel findings include the co-regulation of both Th1 and Th2 responses by iNOS during S. japonicum infection as well as varied roles for iNOS in first suppressing and then enhancing fibrosis in a pulmonary granuloma model. However, the paper fails to present experiments that establish direct causal links between iNOS and its necessity for cellular immunity that mediates protection against S. japonicum. I am further concerned that throughout the manuscript, the authors draw several key conclusions that are either vaguely worded or unsupported by their data. While I describe major additional experiments that could be performed to establish the causal links that the authors claim, I think that for the authors’ interpretations to align more precisely with the data they present will require a major reorganization and rewrite of the manuscript. For those reasons, I recommend to reject the manuscript at this time.

PLOS authors have the option to publish the peer review history of their article (what does this mean?). If published, this will include your full peer review and any attached files.

Reviewer #1: No

Reviewer #2: No
---

## [Decision Letter · Decision Letter 1]

10 Feb 2022

Dear Dr. Zhao-Rong Lun,

Thank you very much for submitting your manuscript "iNOS is essential to maintain host immune activity against Schistosoma japonicum infection in rats" for consideration at PLOS Neglected Tropical Diseases. As with all papers reviewed by the journal, your manuscript was reviewed by members of the editorial board and by several independent reviewers. The reviewers appreciated the attention to an important topic. Based on the reviews, we are likely to accept this manuscript for publication, providing that you modify the manuscript according to the review recommendations. 

Sincerely,

Alessandra Morassutti, PhD

Associate Editor

Walderez Dutra

Deputy Editor

Reviewer's Responses to Questions

**Key Review Criteria Required for Acceptance?**

**Methods**

-Are the objectives of the study clearly articulated with a clear testable hypothesis stated?

-Is the study design appropriate to address the stated objectives?

-Is the population clearly described and appropriate for the hypothesis being tested?

-Is the sample size sufficient to ensure adequate power to address the hypothesis being tested?

-Were correct statistical analysis used to support conclusions?

-Are there concerns about ethical or regulatory requirements being met?

Reviewer #1: No comment.

Reviewer #2: All my methodological critiques have been addressed, particularly with the experiments employing adoptive transfer of WT macrophages.

**Results**

-Does the analysis presented match the analysis plan?

-Are the results clearly and completely presented?

-Are the figures (Tables, Images) of sufficient quality for clarity?

Reviewer #1: No comment.

Reviewer #2: See comments under "Conclusions".

**Conclusions**

-Are the conclusions supported by the data presented?

-Are the limitations of analysis clearly described?

-Do the authors discuss how these data can be helpful to advance our understanding of the topic under study?

-Is public health relevance addressed?

Reviewer #1: No comment.

Reviewer #2: Several related concerns regarding reporting of results and interpretation of Figure 4:

Lines 262-264: “As shown in Fig.4A and 4B, the frequency of CD4+ T cells producing IL-4 was slightly increased in the blood (P<0.05) of naïve KO rats, compared with naïve WT rats.” Fig 4B shows no significant difference in %IL-4+ CD4 T cells between WT naïve and KO naïve. Please correct this text.

Lines 266-270: “However, although the frequency of IL-4-producing CD4+ T cells slightly increased in infected KO rats, when compared to naïve KO rats, no significant differences were found in LN and spleens (P>0.05) although these were lower than those found in the infected WT rats.” The phrasing here is unclear – Fig 4B shows that in KO animals, only blood showed a significant increase between naïve and infected, however this is not specified in the text. Also when the authors state “these were lower than those found in the infected WT rats” it seems they are saying that the *magnitude* of increase between naïve and infected is higher in the WT than in the KO animals. Please clarify the language reporting these results.

Line 270: “The results suggest an impaired Th2 response in infected iNOS-KO rats.” I do not think the results in Fig 4 support this interpretation; although the magnitude of the increase in %IL-4+ or %IFN-gamma between naïve and infected is higher in WT than in KO rats, the actual percentage of IL-4+ or IFN-gamma CD4 T cells is not significantly different between infected WT and infected KO rats (except for IL-4+ in the blood, which decreases, as shown in Fig 4B). So because the % of cytokine positive CD4 T cells is not different between infected WT and infected KO, I don’t think you can argue that these results alone indicated an impaired Th2 (or Th1) response. In contrast, Fig 4E and 4F provide much stronger evidence showing that both Th1 and Th2 responses are impaired in KO animals.

Lines 270-274: “The reduction in the Th2 response did not result from an increased Th1 response because the frequency of CD4+ T cells producing IFN-γ was also decreased in the blood, LN, and spleens of infected KO rats, compared with infected WT rats (P<0.05) (Fig.4C and 4D).” Similar problem as with Lines 266-274 – there are no significant differences between infected WT and infected KO rats in %IFN-g CD4 T cells in blood, LN, or spleen. Naïve animals show lower %IFN-g CD4 Ts in KO vs WT for LN and spleen, but not blood, and the magnitude of increase in %IFN-g CD4 Ts between naïve and infected is smaller for KO vs WT in blood, but is actually higher in KO vs WT in LN and spleen. Please be more precise in accurately reporting these data in the Results section.

Figure 8: I recommend revising the part of the schematic showing that CD3+ and CD4+ T cells decrease with infection when iNOS is present, but that these effects are reversed when iNOS is absent, as this is not supported by the data in Fig 2. In Fig 2B, infection of WT rats does show a non-significant trend towards decreasing T cell %, but this trend does not reverse when KO rats are infected, so I don’t think this warrants inclusion in the summary schematic. In contrast, in Fig 2D in the spleen, CD4 T cells go down when WT rats are infected, but increase when KO rats are infected, so this does fit the current version of the schematic, however this trend does not reach statistical significance for the LN and is not present in the liver, so the schematic should be revised to reflect this accurately.

**Editorial and Data Presentation Modifications?**

Reviewer #1: No comment.

Reviewer #2: In the Financial Disclosure, the authors need to submit a statement explicitly addressing whether sponsors or funders played “any role in the study design, data collection and analysis, decision to publish, or preparation of the manuscript.”

There remain multiple grammatical and typographical errors that will need to be corrected prior to publication.

**Summary and General Comments**

Reviewer #1: No comment.

Reviewer #2: This revised manuscript by Shen et al. employs an iNOS knockout rat model to illustrate the key roles of iNOS and NO signaling in immunopathology and protective immunity to Schistosoma japonicum. The authors’ revisions have substantially improved the quality of the manuscript, particularly by (1) demonstrating that adoptive transfer with WT macrophages partially rescues several of the immune phenotypes, (2) using more precise language in reporting their research findings, and (3) delineating more clearly in their Discussion between interpretations that are firmly supported by the data versus more speculative hypotheses that provide the foundation for future studies. My remaining critiques are comparatively minor and focus on more precise reporting and interpretation of the results in Figure 4, as well as revision of the summary schematic in Figure 8 to more accurately reflect the data on T cells and CD4 T cells reported in Figure 2. With these revisions, coupled with the thorough characterization of the iNOS-KO phenotype in Schistosoma japonicum infection, this has the potential to be an impactful manuscript for studies of mammalian immunity in schistosomiasis. I recommend Minor Revisions before the manuscript is accepted for publication.

PLOS authors have the option to publish the peer review history of their article (what does this mean?). If published, this will include your full peer review and any attached files.

Reviewer #1: No

Reviewer #2: No

Figure Files:

Data Requirements:

Reproducibility:

References

---

## [Decision Letter · Decision Letter 2]

28 Mar 2022

Dear Dr Zhao-Rong, 

Thank you very much for submitting your manuscript "iNOS is essential to maintain host immune activity against Schistosoma japonicum infection in rats" for consideration at PLOS Neglected Tropical Diseases. As with all papers reviewed by the journal, your manuscript was reviewed by members of the editorial board and by several independent reviewers. The reviewers appreciated the attention to an important topic. Based on the reviews, we are likely to accept this manuscript for publication, providing that you modify the manuscript according to the review recommendations. 

Sincerely,

Alessandra Morassutti, PhD

Associate Editor

Walderez Dutra

Deputy Editor

Reviewer's Responses to Questions

**Key Review Criteria Required for Acceptance?**

**Methods**

-Are the objectives of the study clearly articulated with a clear testable hypothesis stated?

-Is the study design appropriate to address the stated objectives?

-Is the population clearly described and appropriate for the hypothesis being tested?

-Is the sample size sufficient to ensure adequate power to address the hypothesis being tested?

-Were correct statistical analysis used to support conclusions?

-Are there concerns about ethical or regulatory requirements being met?

Reviewer #2: (No Response)

Reviewer #3: (No Response)

Reviewer #4: (No Response)

**Results**

-Does the analysis presented match the analysis plan?

-Are the results clearly and completely presented?

-Are the figures (Tables, Images) of sufficient quality for clarity?

Reviewer #2: (No Response)

Reviewer #3: (No Response)

Reviewer #4: (No Response)

**Conclusions**

-Are the conclusions supported by the data presented?

-Are the limitations of analysis clearly described?

-Do the authors discuss how these data can be helpful to advance our understanding of the topic under study?

-Is public health relevance addressed?

Reviewer #2: (No Response)

Reviewer #3: (No Response)

Reviewer #4: (No Response)

**Editorial and Data Presentation Modifications?**

Reviewer #2: (No Response)

Reviewer #3: (No Response)

Reviewer #4: The authors have presented massive amount of work and is highly appreciated. 

Here are my observations:

• Suggestion for Title:

“Knocking-out iNOS in rats disintegrates a balanced and protective Th1-Th2 immune response against Schistosoma japonicum, exacerbating acute pro-inflammatory response leading to host mortality”

• Flow cytometric data displayed in Fig 2 B and 2 D histograms, is based on the percentage of different cell populations. I wish if the numbers of different cell populations can also be displayed in similar histograms to have an overview- how many times the immune cells have multiplied? Whether the numbers of these different cell populations, are also reflecting the same scenario as evident from their percentages? This could be really challenging but is not impossible. The starting volume of blood sample, the starting weight of liver, lymph nodes and spleen samples- can be greatly helpful in these calculations, to reach the overall cell numbers in the peripheral circulation as well as in different organs.

• Line 449-450: Please delete “in” [CD4+ T cell-mediated Th1 and Th2 responses play an important role in against infection and immunopathogenesis of schistosomiasis]

• Line 654: Please correct the sign of micron.

**Summary and General Comments**

Reviewer #2: (No Response)

Reviewer #3: This manuscript may be suitable for acceptance as the comments of the previous reviewers have been well addressed. My only suggestion is that the resolution of the figures, especially the histo-immunostaining pictures, needs to be improved to show the area indicated by the arrow more clearly.

Reviewer #4: (No Response)

PLOS authors have the option to publish the peer review history of their article (what does this mean?). If published, this will include your full peer review and any attached files.

Reviewer #2: No

Reviewer #3: No

Reviewer #4: Yes: Dr Haroon AKBAR, Faculty of Veterinary Science, University of Veterinary and Animal Sciences, Lahore, Pakistan

Figure Files:

Data Requirements:

Reproducibility:

References

---

## [Editor Report · Decision Letter 3]

8 Apr 2022

Dear Dr Zhao-Rong Lun, We are pleased to inform you that your manuscript 'iNOS is essential to maintain a protective Th1/Th2 response and the production of cytokines/chemokines against Schistosoma japonicum infection in rats' has been provisionally accepted for publication in PLOS Neglected Tropical Diseases.

Best regards,

Alessandra Morassutti, PhD

Associate Editor

Walderez Dutra

Deputy Editor

---

## [Editor Report · Acceptance letter]

3 May 2022

Dear Dr Lun,

We are delighted to inform you that your manuscript, "iNOS is essential to maintain a protective Th1/Th2 response and the production of cytokines/chemokines against Schistosoma japonicum infection in rats," has been formally accepted for publication in PLOS Neglected Tropical Diseases.

Best regards,

Shaden Kamhawi

co-Editor-in-Chief

Paul Brindley

co-Editor-in-Chief
